# Bioengineered niches that recreate physiological extracellular matrix organisation to support long-term haematopoietic stem cells

Hannah Donnelly [1], Ewan Ross[1], Yinbo Xiao[1], Rio Hermantara [2], Aqeel F. Taqi[2], W. Sebastian Doherty-Boyd [1], Jennifer Cassels[3], Penelope. M. Tsimbouri [1], Karen M. Dunn [3], Jodie Hay [3], Annie Cheng[4], R. M. Dominic Meek[5], Nikhil Jain[6], Christopher West[7], Helen Wheadon [3], Alison M. Michie [3], Bruno Peault[7], Adam G. West[2], Manuel Salmeron-Sanchez [4] ✉ & Matthew J. Dalby [1] ✉

Long-term reconstituting haematopoietic stem cells (LT-HSCs) are used to treat blood disorders via stem cell transplantation. The very low abundance of LT-HSCs and their rapid differentiation during in vitro culture hinders their clinical utility. Previous developments using stromal feeder layers, defined media cocktails, and bioengineering have enabled HSC expansion in culture, but of mostly short-term HSCs and progenitor populations at the expense of naive LT-HSCs. Here, we report the creation of a bioengineered LT-HSC maintenance niche that recreates physiological extracellular matrix organisation, using soft collagen type-I hydrogels to drive nestin expression in perivascular stromal cells (PerSCs). We demonstrate that nestin, which is expressed by HSC-supportive bone marrow stromal cells, is cytoprotective and, via regulation of metabolism, is important for HIF-1α expression in PerSCs. When CD34[+ve] HSCs were added to the bioengineered niches comprising nestin/HIF-1α expressing PerSCs, LT-HSC numbers were maintained with normal clonal and in vivo reconstitution potential, without media supplementation. We provide proof-of-concept that our bioengineered niches can support the survival of CRISPR edited HSCs. Successful editing of LT-HSCs ex vivo can have potential impact on the treatment of blood disorders.

Haematopoietic diseases are a major healthcare burden, costing the UK economy £4B pa[1], with leukaemia therapy typically requiring multiple courses of intensive chemotherapy, often followed by allogeneic haematopoietic stem cell (HSC) transplant (alloSCT). AlloSCT transplantation replaces a patients hematopoietic cells through myeloablation of host bone marrow (BM), followed by reconstitution with transplanted donor cells to restore normal immune function[2]. Given that alloSCT has been demonstrated to be an increasingly efficacious therapy for not only leukaemias but also for primary immunodeficiencies and autoimmune conditions, there is an increasing demand for HSC transplantation and for matched donors[3], particularly in the context of an aging and ethnically diverse population. However, we currently cannot meet the clinical demand as there are no defined and regulated expansion methodologies for BM HSCs, meaning that we are reliant on donation. Furthermore, engraftment of HSCs can be limited following alloSCT, leading to prolonged haemopoietic

recovery time post-transplantation, which is a major cause of transplant failure and patient mortality[2].

Long-term (LT)-HSCs are critically important for therapy as they engraft to repopulate the blood system following alloSCT[4]. However, LT-HSCs rarely divide and represent just 0.01–0.04% of the mononuclear BM population[5]. They rapidly differentiate towards short-term (ST)-HSCs and progenitor cells upon their culture in vitro, which provide only temporary engraftment and myeloid/lymphoid progenitors[4,5]. Therefore, maintaining and, ultimately, expanding LT-HSCs ex vivo has become a major focus of research. Several culture systems, such as Dexter cultures[6], and the use of cytokines[7] and small molecules[8], have successfully expanded heterogenous CD34+ HSCs in vitro but with the concomitant loss of naive LT-HSCs, producing populations with clonal heterogeneity[9]. Subsequent transplantation studies and phase I/II clinical trials (e.g., with the small molecule Stem Reginin-1) have shown that most engrafted cells are ST-HSCs and their subsequent progenitors[10].

The inability to maintain LT-HSCs, or to support their self-renewal, in these systems, is likely to be related to the lack of the many physical and functional signalling components of the highly complex BM microenvironment, where the expansion and differentiation of HSCs and other hematopoietic cells are regulated[11]. To maintain LT-HSCs ex vivo, there needs to be defined conditions that allow activation and proliferation without inducing differentiation to ST-HSC and subsequent committed progenitors[4]. This has proven to be impossible to achieve in a heterogenous BM cell population, especially when a single parameter, such as one signalling pathway, is targeted with small molecules[8]. Multiple cell types also interact in the BM and co-operate with each other through cell-cell adhesion and paracrine signals, such as mesenchymal stromal cells (MSCs) and perivascular stromal cells (PerSCs)[12–16]. Other critical cellular factors in the BM stem cell niche are C-X-C motif chemokine 12 (CXCL12, also known as stromal cell-derived factor 1 (SDF1)) abundant reticular (CAR) cells[17], and bone-forming osteoblasts[15,16] (Fig. 1). All are implicated in LT-HSC maintenance, while nestin+ve PerSCs are specifically implicated in LT-HSC self-renewal in murine studies[12,13,18]. This observation of nestin+ve stromal cells being important to LT-HSC phenotype is interesting as soft materials, such as hydrogels, have been implicated in the expression of nestin in MSCs[19]. While originally this was ascribed to a neural phenotype[19], we hypothesise this seminal observation can be important in engineering a niche-stromal phenotype and that stromal nestin expression will provide key niche functionalities.

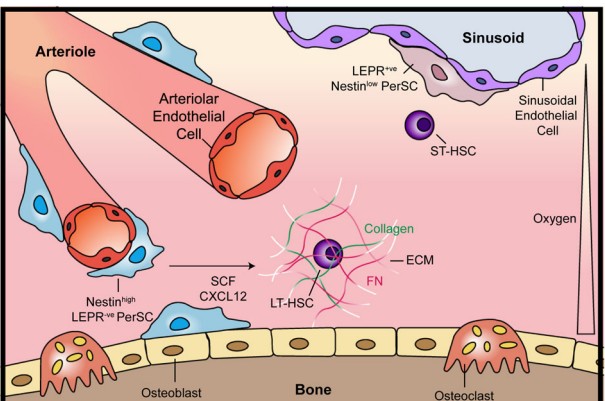

**Fig. 1 | Schematic of the bone marrow (BM) niche microenvironment.** Long-term HSCs (LT-HSC) and Nestin^high/LEPR^−ve Perivascular stem cells (PerSCs) reside close to the endosteum and the oxygenated arterioles that perfuse the endosteal niche, whereas short-term HSCs (ST-HSCs) and LEPR^+ve/Nestin^low PerSCs are in close proximity to the sinusoidal vessels that carry deoxygenated blood away from the BM. The BM extracellular matrix (ECM) is a low-stiffness network comprised primarily of collagen and fibronectin (FN). Modified with permissions from ref. 83.

Indeed, the BM itself is a soft material, comprising extracellular matrix (ECM) with a Young's modulus of $1–10^4$ Pa[20], that interfaces with hard materials, such as the bone lining (endosteal) surfaces[21] and the arterioles and sinusoidal capillaries, to which PerSCs adhere, affecting their mechanotransductive signalling and phenotype[13,22]. Growth factors (GFs) and cytokines interact with ECM's structural proteins, such as fibronectin (FN), facilitating synergistic signalling between ECM-adhering integrin receptors and GF receptors, further influencing the stromal phenotype[23]. The BM niche also features hypoxic environments[24]. The endosteal niche was traditionally considered to be highly hypoxic, but it has been shown that the endosteal surface is perfused with a network of oxygenated arterioles[13,24]. By contrast, the central cavity is filled with more hypoxic sinusoids[13,24]. Hypoxic conditions can influence cell phenotypes, with hypoxic culture conditions reducing stromal cell proliferation and maintaining their stem cell phenotype[25]. Indeed, hypoxia-inducible factor (HIF) expression by stromal cells is believed to be important for their HSC regulatory capacity[26].

We propose that multiple elements of the BM's complexity need be recapitulated in vitro to maintain LT-HSCs. Numerous in vitro models of the BM niche are currently in development. Some integrate multiple types of support cells[18,27], 3D hydrogels[28,29] and scaffolds[30], while others employ perfusion cultures[31], and combine different approaches on organ-on-chip devices[32]. While these emerging models inform our understanding of the human BM during injury or neoplasia[32], and of CD34+ve HSC expansion[18,27], they have their limitations. Many use murine CD34+ve cells, despite the notable differences between human and murine HSC regulation[33], others measure expansion based on CD34+ve numbers rather than on the LT-HSC population, while some models are complex or hard to use or recover cells from[4]. To our knowledge, there is no reported system that focuses on enhancing human LT-HSC maintenance in a simple-to-use, practical, system. Such a system could inform on both maintenance of LT-HSCs for therapeutic use as well as providing a platform to test novel drugs or gene editing of HSCs in vitro, that can offer novel therapeutic cell-based therapies to treat malignant and non-malignant blood disorders.

In this study, we employ bioengineering approaches in order to direct the physiology of the stromal feeder layer and enhance its ability to support and maintain quiescent LT-HSCs in culture without unwanted drift towards non-engrafting ST-HSCs and progenitor cells. In this study, we define LT-HSCs in vitro by survival in a long-term culture initiating cell assay (LTC-IC) and in vivo by engraftment post 3-month. For the bioengineered niche, we employ a polymeric surface that controls fibronectin (FN), a major component of BM ECM[34], fibril formation allowing both an adhesive and direct GF signal to stromal cells[35,36], as well as a soft material to induce nestin expression to stimulate the stromal HSC-supportive phenotype[12,13,18]. Our findings show that nestin expression, linked to changes in respiration, are central to enhanced LT-HSC maintenance. We further demonstrate that our bioengineered niches have the potential to support improved survival of CRISPR edited HSCs in this clinically important population.

## Results
### Recapitulating the endosteal surface in vitro
The BM niche has different putative regions with different roles in HSC regulation. The sinusoidal (central marrow cavity) niche is associated with sinusoidal capillaries and is more hypoxic relative to other BM regions[24], and self-renewing ST-HSCs are considered to reside there[14]. The endosteal surface, which is in close proximity to a large network of oxygenated arterioles, is believed to retain a more quiescent pool of LT-HSCs[12,13,24,37,38]. The BM also has a dense ECM network, the constituents of which are exchanged between the endosteal and sinusoidal niches (Fig. 1). At the endosteal surface, the ECM is stiffer and FN-rich. Towards the sinusoidal niches in the central BM cavity, the ECM

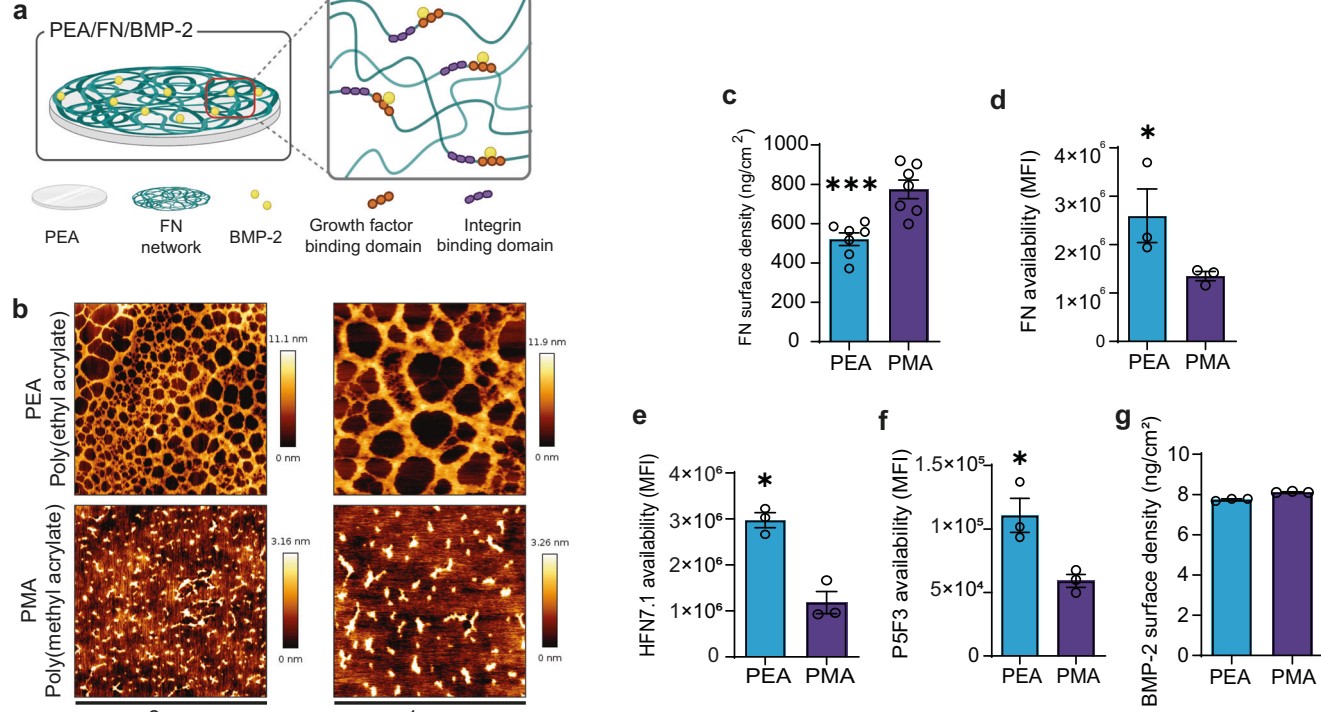

**Fig. 2 | Control of the protein interface using acrylate polymers. a** Schematic of the PEA system. **b** AFM height images showing fibronectin (FN) spontaneously forms networks on PEA surfaces but not on PMA. **c** Surface density of FN on PEA and PMA surfaces coated with 20 µg/mL FN solution, showing that more FN adsorbs to PMA ($n = 7$ material replicates, $p = 0.0006$). The availability of **d** total FN ($p = 0.05$); **e** FN HFN7.1 domain (RGD-binding) ($p = 0.05$); and **f** P5F3 domain (growth factor (GF)-binding) ($p = 0.05$), was assessed using in-cell western analysis using antibodies against each domain. Data shows the increased availability of RGD- and GF-binding FN domains on PEA; MFI mean fluorescent intensity. **g** Surface density of BMP-2 on PEA-FN and PMA-FN, as assessed by ELISA. 50 ng/mL of BMP-2 was loaded onto the surface, leading to ~8 ng/cm² of BMP-2 sequestered on both PEA and PMA (**d**–**g** $n = 3$). **c**–**g** Graphs show mean ± SEM, *$p < 0.05$, ***$p < 0.001$ determined by one-tailed student $t$ test (Mann–Whitney), $n = 3$ material replicates. Source data are provided as a Source Data file. **a** created with BioRender.com released under a Creative Commons Attribution-NonCommercial-NoDerivs 4.0 International license (https://creativecommons.org/licenses/by-nc-nd/4.0/deed.en).

becomes progressively less stiff and more collagen- and laminin-rich[39,40]. The conformation of ECM proteins, as well as their type and density, all strongly influence BM cell interactions[41].

Our previous work has shown that fibrillar FN can efficiently bind and present BMP-2 (bone morphogenetic protein 2, an osteogenic GF) to promote osteogenesis in MSCs in vitro[35] and fracture healing in vivo[36]. We have adapted these findings here to recreate the ECM properties of the more quiescent endosteal niche. Poly(ethyl acrylate) (PEA) leads to the spontaneous unfolding of FN (Fig. 2a), as visualised with atomic force microscopy (AFM) (Fig. 2b, Supplementary Fig. 1a). This cell-driven unfolding process leads to the exposure of key binding domains in the FN molecule[42]. Chemically similar poly(methyl acrylate) (PMA) is used as a reference polymer here, on which FN is adsorbed and maintained in a globular conformation (Fig. 2b, Supplementary Fig. 1a). PMA behaves similarly to PEA in terms of surface wettability and stiffness[43], but with increased adsorption of FN (Fig. 2c). The fluorescent detection of antibodies that recognise total FN (Fig. 2d, Supplementary Fig. 1b), FN's arginine-glycine-aspartic acid (RGD)-containing HFN7.1 integrin-binding domain (Fig. 2e), and its P5F3 GF-binding domain (Fig. 2f) demonstrates the significantly increased availability of these FN domains when absorbed on PEA, relative to PMA. The P5F3 domain of FN, in particular, has high binding affinity for several GF families[44]. Here, we show that exogenous BMP-2 binds to PEA-FN (Fig. 2g) allowing us to present low levels of this endosteal-related GF to cells. Although similar amounts of BMP-2 also bind to PMA, our previous work has shown that BMP-2 does not co-localise with FN on PMA, leading to its faster release[35,36]. Thus, in this study, we use PEA to present FN and BMP-2 to recreate the milieu of the

endosteal surface that supports HSC quiescence (maintenance) in our BM model.

## Control of PerSC phenotype

The stromal population of the BM niche is highly heterogeneous[11]. Reported expression of markers, such as nestin, neural-glial antigen 2 (NG-2) and leptin receptor (LEPR), indicate that PerSCs and MSCs have a phenotype that supports HSC regulation[12–14]. Here, we focus on the LT-HSC regulating PerSCs that line the endosteal arterioles[12,13,18], and on nestin. Nestin is a cytoskeletal intermediate filament (IF) protein traditionally associated with neural progenitor cells[45]. In vivo murine BM studies have shown that nestin+ve PerSCs express high levels of factors involved in HSC maintenance, such as stem cell factor (SCF)[38]. Previous seminal work has also shown that stromal cells express nestin in response to soft materials[19]. We, therefore, used low-stiffness collagen type I hydrogels, a major constituent of the BM ECM[39,46], to study the role of nestin expression in human PerSCs and to investigate how this influences their interactions with HSCs in vitro. We note that PerSCs have similar multipotency to MSCs (Supplementary Fig. 2).

The collagen gels, with a Young's modulus of ~80 Pa to mimic BM HSC niche stiffness[20], were overlayed onto PerSCs cultured on PEA/FN/BMP-2 surfaces (Fig. 3a, b, Supplementary Fig. 3). We also used PEA/FN/BMP-2 with a lowered oxygen tension (1% O₂, hypoxia), since hypoxia is considered to be important for maintaining the stromal HSC supporting phenotype in the BM environment[24] (Fig. 3a). We observed in our bioengineered PEA/FN/BMP-2 niches with hydrogels (denoted as +gel) that nestin was significantly upregulated by day 7 in PerSCs (Fig. 3c), with enhanced nestin expression continuing to day 14

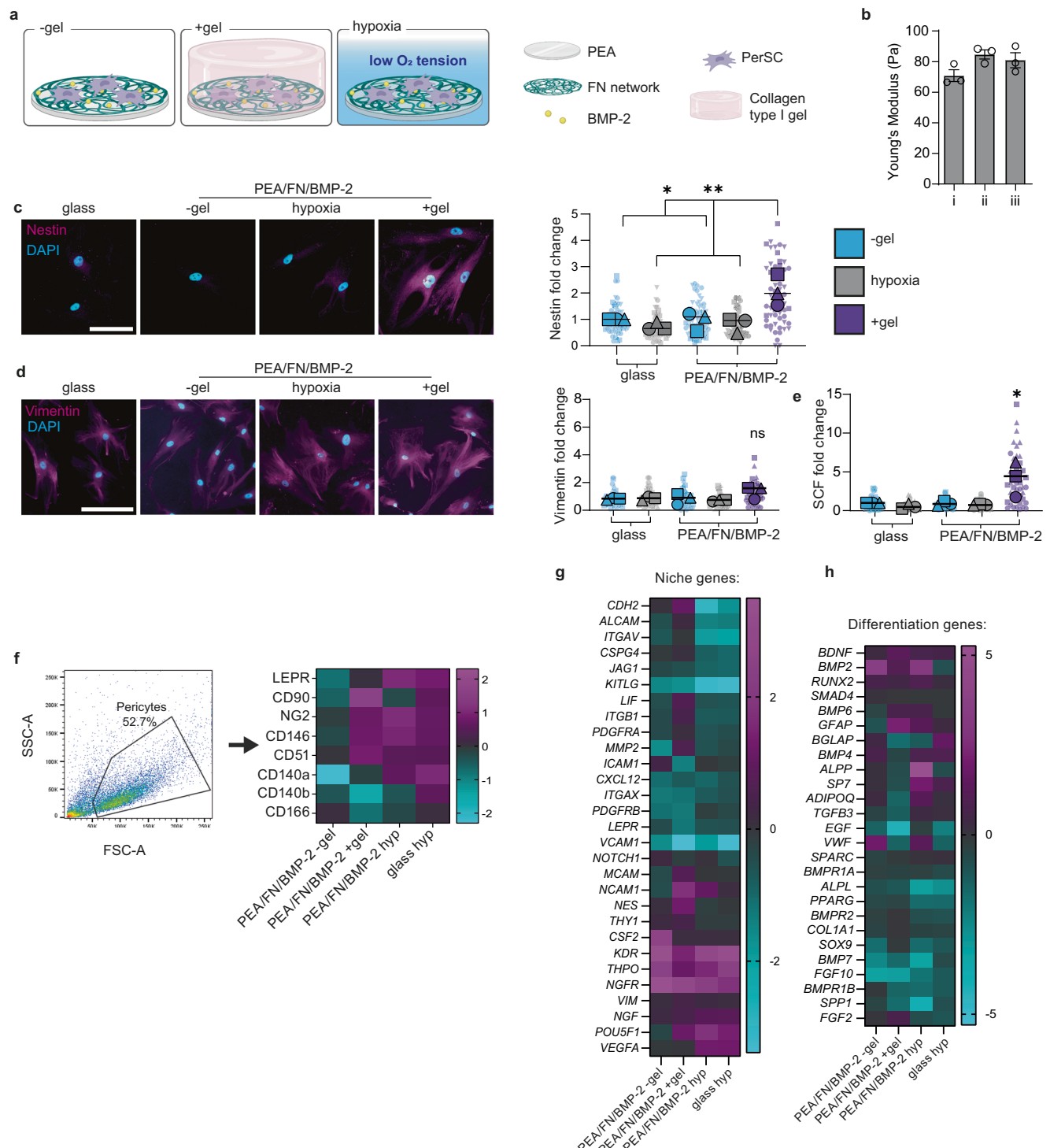

(Supplementary Fig. 4a). By contrast, nestin was expressed at a basal level in PEA/FN/BMP-2 niches without hydrogels (denoted as −gel), in the PEA/FN/BMP-2 hypoxic niches and in PMA/FN/BMP-2 niches with and without gels (Supplementary Fig. 4a). We used PerSCs on glass coverslips as a culture control since they do not attach to highly hydrophobic PEA without a protein interface (Fig. 3c–f). Thus, nestin expression was induced only when the gel was included on PEA-unfolded FN. Nestin is a type IV IF protein incapable of self-polymerisation. It, therefore, co-polymerises with other type II and IV IFs[45]. We therefore examined vimentin expression, as an IF protein expressed in mesenchymal-lineage cells, and found no significant differences in vimentin expression or its morphology (Fig. 3d).

Next, PerSCs were treated with brefeldin A for the final 24 h of culture to prevent protein export and SCF levels were monitored by immunofluorescence. SCF production correlated with the observed increase in nestin expression in PEA +gel niches (Fig. 3e, Supplementary Fig. 4b, c). CXCL12 expression was also detected in PerSCs on all substrates, with no significant differences between the bioengineered models (Supplementary Fig. 4d).

Phenotypic analysis by flow cytometry of the different environments showed that PerSCs in +gel niches expressed important stromal niche markers CD90, NG2, CD146 and CD51, but that expression was lost in −gel niches[13,16,18] (Fig. 3f, Supplementary Fig. 5). Interestingly, the introduction of hypoxia also led to expression of these markers, and

**Fig. 3 | Investigating bioengineered niches that support BM stromal phenotypes. a** Schematic of niche model set-up. **b** Young's modulus of collagen type I gels used in +gel niches. i–iii represent 3 independent batches of gels formulated. The average Young's modulus was 78.8 Pa; no statistical significance was observed, as determined by one-way ANOVA followed by Bonferroni multiple comparison. Graph shows mean + SEM for 3 measurements per batch. **c** Representative images of changes in nestin expression evaluated by immunofluorescence at day 7 show increased expression in PEA/FN/BMP-2 +gel. Scale bar is 100 µm, magenta = nestin, cyan = DAPI. **d** Representative images of vimentin expression, as evaluated by immunofluorescence microscopy on day 7, shows no significant difference between models, scale bar is 100 µm, magenta = vimentin, cyan = DAPI. **e** SCF production in bioengineered models, in which PerSCs were cultured for 14 days, with brefeldin A added for the final 24 h to inhibit intracellular protein transport. Representative images in Supplementary Fig. 5c. Graphs in **c**–**e** show mean integrated intensity of marker as fold change to glass control; −gel = blue, +gel = purple,

hypoxia = grey. Each point represents an individual field normalised to cell number. $N = 3$ material replicates, for 3 independent experiments with different donor cells (each donor represented by a different shape), *$p < 0.05$, **$p < 0.001$, ns non-significant, determined by one-way ANOVA followed by Bonferroni's multiple comparison test. **f** Assessment of PerSC phenotype after 14 days culture in niche models. Heatmap shows fold change over control (glass) of median fluorescent values obtained by flow cytometry from 6 pooled material replicates from $n = 3$ independent experiments with different donor cells. At least 5000 cells were collected per sample. **g**, **h** RNA-seq was performed on PerSCs from niche models after 7 days, and fold changes in gene expression relative to glass control are shown as a heatmap. $N = 6$ pooled material replicates from 1 donor. Source data are provided as a Source Data file. **a** created with BioRender.com released under a Creative Commons Attribution-NonCommercial-NoDerivs 4.0 International license (https://creativecommons.org/licenses/by-nc-nd/4.0/deed.en).

also increased expression of LEPR. LEPR is a marker of stromal cells that line hypoxic sinusoids, and supports our attempt to recreate an arteriolar/endosteal niche in which stromal cells are reported to be nestin$^{+ve}$/LEPR$^{−ve}$[12,14,38] (Fig. 3f, Supplementary Fig. 5).

Next, we performed RNA-seq on PerSCs cultured in the PEA/FN/BMP-2 −gel, +gel and hypoxic niches. RNA-seq identified transcripts related to niche (Fig. 3g) or regulation of differentiation (Fig. 3h). Importantly, cells from the +gel niche had an up-regulation of niche genes and a concomitant downregulation of differentiation genes, compared to control and other conditions (−gel and hypoxia) (Fig. 3g, h). The known functions of these genes are listed in supplementary Tables 1 and 2.

Using principal component analysis (PCA), the −gel/+gel/hypoxic conditions were clearly distinguishable from each other by RNA-seq (Supplementary Fig. 6a). Analysis of significant gene changes revealed that most unique changes occurred in the +gel niche, and that the +gel and hypoxic models had more changes in common, relative to the −gel model (Fig. 4a). Gene ontology (GO) clustering analysis indicates that a range of changes occurred in response to the different engineered ECM environments (−gel / +gel) (Supplementary Fig. 6b). Changes in integrins, adhesion molecules, ECM interactions, cytoskeleton and to the BMP-2 milieu, were evident in response to these ECM changes, as were changes to members of the canonical BMP signalling family, including TGFβs and SMADs (Fig. 4b). Clustering analysis of the four top networks of actin cytoskeleton, cell adhesion and ECM, illustrated that in the +gel niches, the PerSCs had decreased levels of cytoskeletal transcripts and increased levels of cell adhesion and ECM transcripts, compared to the −gel models (Fig. 4c). The hypoxic niche, however, both upregulated and downregulated ECM transcripts, and strongly repressed adhesion and actin related transcripts, relative to the other models (Fig. 4c, Supplementary Fig. 6c).

Another feature of the naive stromal stem cell phenotype is the ability to modulate the immune system[47]. We therefore compared the expression profiles of inflammation-related marker genes across the three different niches. Several markers up-regulated in PerSCs in the +gel niche, compared to the −gel model, are implicated in immunomodulatory properties of the stromal phenotype, including interleukin 6 (IL-6)[48], prostaglandin-endoperoxide synthase 2 (PTGS2), prostaglandin E2 receptor 2 (PTGER2)[49], CD44, tumour necrosis factor α-induced protein 3 (TNFAIP3)[50] and ephrin type-A receptor 2 (EPHA2)[51] (Fig. 4d). PerSCs in the hypoxic and −gel models showed reduced expression of these immunomodulatory transcripts, with little difference between them, whereas +gel niches led to enhanced expression of these transcripts when compared to both −gel and hypoxic niches (Fig. 4d). Together, these findings help demonstrate that a naive PerSC phenotype is retained in the +gel niches.

## Metabolic regulation in the BM niche

In order to elucidate further the role of hypoxia in the BM niche, we investigated the role of hypoxia and stiffness on stromal niche phenotypes by using our modular platform to compare the effects of hypoxia in our bioengineered models. HIF1α is a master regulator of metabolism. It regulates both glycolysis and oxidative phosphorylation, including anaerobic glycolysis[52]. As such, we evaluated the expression pattern of HIF1α in PerSCs in our bioengineered niches after 3, 7 and 14 days of culture. HIF1α is constitutively produced but is hydroxylated in normoxia which inhibits its transcriptional activity and targets it for ubiquitination and degradation in most cells[52]. However, in hypoxia, it becomes stable, translocates to the nucleus and activates the transcription of hypoxia-responsive genes. We, therefore, chose to investigate nuclear expression of HIF1α (Fig. 5a, Supplementary Fig. 7a–c). By this measure, all three bioengineered models led to increased nuclear expression of HIF1α protein in PerSCs. Interestingly, however, cells in the +gel niches displayed significantly more nuclear HIF1α, expression compared to the −gel and hypoxic models at all time points; the +gel niche also supported PerSCs with the most niche-like phenotype (Fig. 3c–h).

Although some hypoxic response genes are up-regulated in PerSCs in the +gel niches (Supplementary Fig. 6c), our investigation of HIF1α downstream targets, such as of VEGF (Fig. 5b), lactate and lactate dehydrogenase (LDH) (Supplementary Fig. 7d, e), did not reveal an increase in expression. To determine if the PerSCs in the +gel niches were experiencing low oxygen tension, we next used pimonidazole, a hypoxia probe. The probe revealed that while PerSCs cultured in the low oxygen tension models are clearly hypoxic, cells in the −gel models and +gel niches were not (Fig. 5c). This indicates that PerSCs in the +gel niche that were expressing nuclear HIF1α were not doing so in response to lowered oxygen tension.

To identify changes in the metabolism of PerSCs in the niche models, liquid chromatography-mass spectrometry (LC-MS) was performed to identify alterations in the metabolome at 7 and 14 days of culture. PCA of all metabolites demonstrated that all three culture conditions clustered with no overlap in PC 1 vs PC 2, and with some overlap between +gel and hypoxic niches at day 14 in PC 2 vs PC 3. (Fig. 5d, Supplementary Fig. 8). Then, focussing only on glycolysis and TCA cycle metabolites (Fig. 5e), revealed an expected increase in clusters of glycolytic metabolites in the hypoxic model, as well as the upregulation of TCA-associated metabolites in the −gel model, as would be expected with active differentiation[53]. Interestingly, for the +gel niche, we observed a general downregulation of glycolysis and oxidative phosphorylation rather than an increase in glycolysis (Fig. 5e).

HIF activity in non-hypoxic conditions is a distinguishing feature of the Warburg effect, in which cancer cells switch to rely on oxidative glycolysis to generate energy[54]. The metabolic profile of

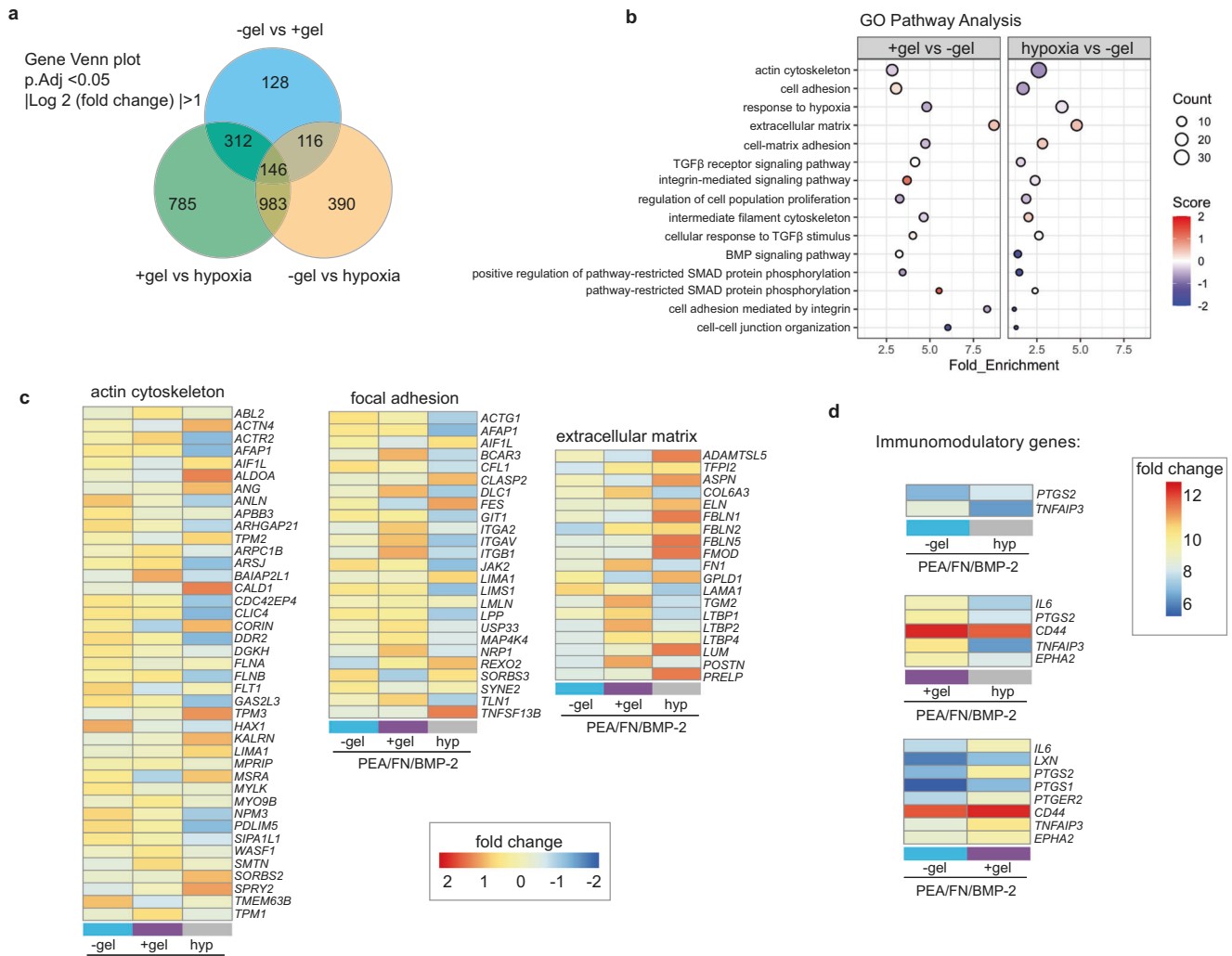

**Fig. 4 | Investigating bioengineered niches that support PerSC phenotype.**
RNA-seq was performed on PerSCs isolated from the bioengineered models at day 7. **a** Venn plot shows differentially expressed genes (based on passing a threshold of Wald test adjusted *p* value < 0.05 and log2 fold change > 1). **b** GO (gene ontology) enrichment analysis for differentially expressed transcripts based on false discovery rate (FDR) ≤ 0.05 linked to pathways involved in ECM, cell adhesion and cytoskeleton. Results indicate that the cells differentially re-organise ECM-relevant gene expression in response to the +gel niche or hypoxic environments. Differentially expressed genes in each of these pathways were extracted, and z scored

changes are shown in the heatmaps. **c** Expression profiles of genes involved in ECM, cell adhesion, and cytoskeleton. In the hypoxic model, ECM-related transcripts are increased while adhesion- and cytoskeleton-related transcripts are decreased. In the +gel niche, ECM and adhesion-related transcripts are more subtly increased while cytoskeleton transcripts are decreased. **d** Immunomodulatory transcript expression, showing similar expression profiles in the hypoxic and −gel models, and up-regulated transcripts in the +gel niches. For RNA-seq, *n* = 3 6 pooled material replicates from 1 donor.

PerSCs cultured in the +gel niche appears to be more quiescent than glycolytic, however. This finding was confirmed by tracing the conversion of C[13] labelled glucose using LC-MS in PerSCs (Fig. 5f). In the +gel niches, PerSCs still retained significantly more labelled glucose compared to other conditions, indicative of lowered respiration. In the −gel models, significantly less glucose was detected at the end of the analysis, and it was not reduced to pyruvate or lactate but there was an increase in ATP levels, indicative of more normal respiration (Fig. 5e). In hypoxic models, significant levels of C[13] labelled pyruvate and lactate remained in the PerSCs, indicative of anaerobic respiration, as would be expected[54]. Taken together, these observations indicate that PerSCs in the +gel niche are not responding as if in hypoxic conditions, but are attenuating respiration. This is despite the increased expression and prolonged nuclear presence of HIF1α, which may be acting in a BM niche-specific role.

## Nestin is required for metabolic regulation in the niche

Little is known about why nestin is highly expressed in BM niche stromal cells. We have already shown that while nestin levels are increased in +gel niches (Fig. 3c), IFs (as measured using vimentin) are not (Fig. 3d). To confirm this observation, we assessed the incorporation of nestin into IF networks using an antibody against p(Thr316)-nestin. This site, when phosphorylated, drives disassembly of nestin from IF fibres[55]. We observed a significant upregulation of p(Thr316) nestin in PerSCs in the +gel niche, indicating that not all nestin is incorporated into the cytoskeleton. This suggests that the function of nestin in stromal cells in our engineered BM niche does not entirely depend on its structural involvement in the cytoskeleton (Fig. 6a, Supplementary Fig. 9).

Because of our HIF1α observation and because soluble nestin has been linked to cytoprotection in other cell types[55–57], we next investigated two possibilities: (1) that nestin expression is linked to the HIF1α

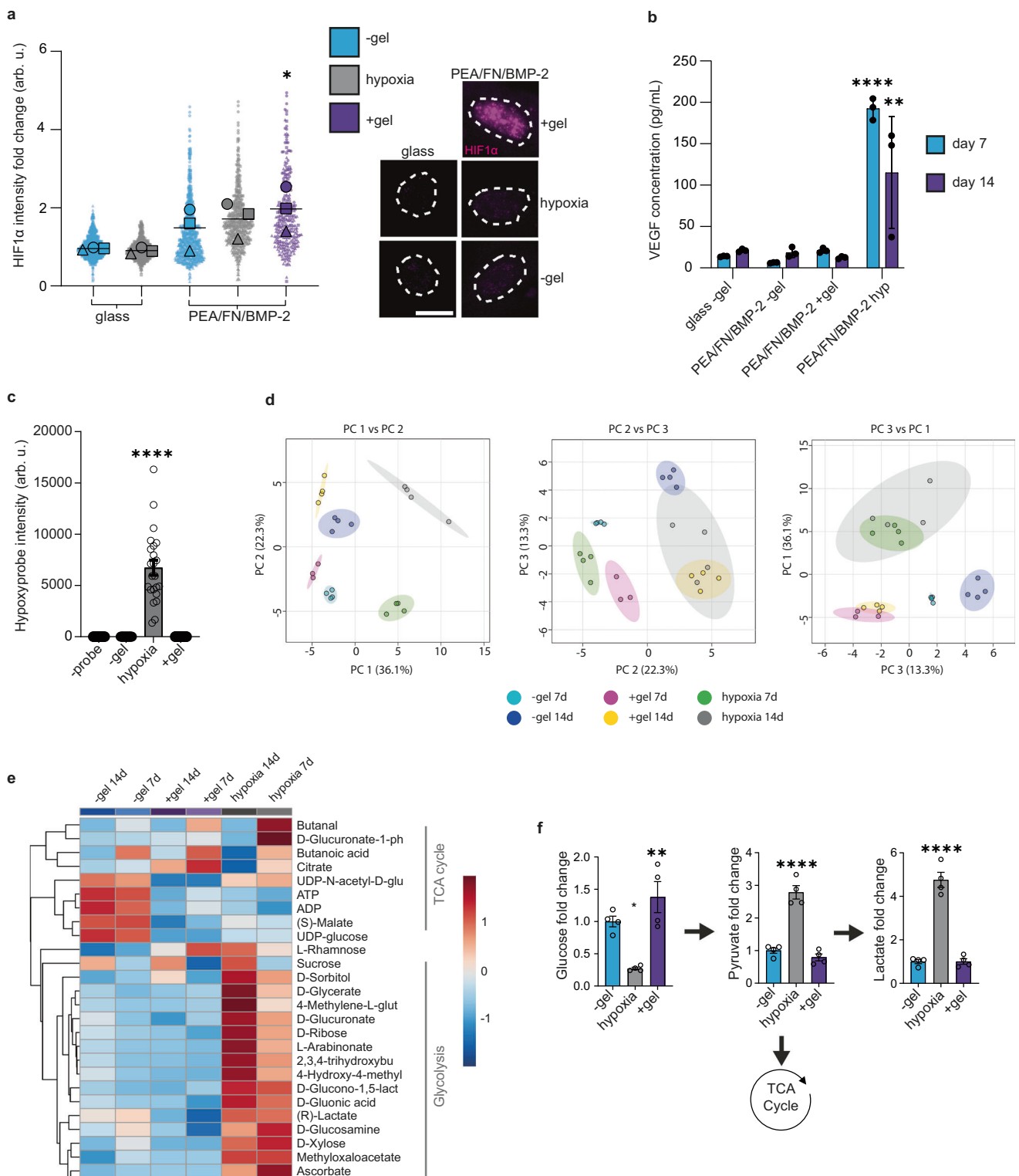

positive PerSC phenotype, and (2) that nestin is linked to cytoprotection. Linking PerSC nestin expression and cytoprotection is perhaps important as naive stromal cells are linked to increased caspase expression (involved in regulation of apoptosis), and enhanced ability to undergo cell death[58] and so perhaps these cells need to be protected by their niche.

To address these questions, nestin expression was reduced by siRNA knock-down in PerSCs to test links between HIF1α expression and cytoprotective effects (Fig. 6b, Supplementary Fig. 10a).

Interestingly, in the +gel niche, nestin knock-down led to a concomitant decrease in the levels of nuclear-localised HIF1α, demonstrating a link between nestin and HIF1α (Fig. 6c, Supplementary Fig. 10b). We note that a recent study indicated that nestin binds to HIF1α to protect it from degradation and showed nestin knockdown leads to loss of HIF1α, as we show here[59]. To evaluate if this link was involved in cytoprotection, oxidative stress and apoptosis were induced in PerSCs by exposure to hydrogen peroxide, and changes in viability were measured using propidium iodide (PI) uptake. In the +gel

**Fig. 5 | Bioengineered niches promote a hypoxic-like metabolic phenotype.**
**a** Nuclear HIF1α levels significantly increased in +gel niches at day 3. Representative images shown, magenta = HIF1α, white = nuclei outline (DAPI), scale bar = 20 μm. $n = 3$; graph shows means from 3 material replicates for 3 independent experiments with different donor cells. Each individual point represents 1 nuclei measurement with shape corresponding to mean; −gel = blue, +gel = purple, hypoxia = grey.
**b** HIF1α downstream analysis; VEGF measured from cell supernatant by ELISA on day 7 (blue) and 14 (purple), graph shows mean ± SEM for $n = 3$ material replicates.
**c** Hypoxia was indicated when PerSCs were cultured in hypoxic but not in the +gel niches. Integrated intensity of hypoxyprobe immunofluorescence shown on graph, bar is mean ± SEM, $n = 4$ material replicates, where each point represents 1× image field normalised to cell number. **d** LC-MS metabolome analysis. Principal component analysis (PCA) of all detectable metabolites in PerSCs in PEA/FN/BMP-2 models at 7 and 14 days. Each point represents 1 replicate. Hypoxic and +gel niches showed similar clustering in PC2. **e** Glycolytic and TCA cycle metabolite profile. Heatmap shows group averages of $\log_{10}$ transformed peak intensities Ward clustered. Glycolysis related metabolites show increased abundance in PEA/FN/BMP-2 hypoxic conditions, whereas TCA cycle metabolites are increased in −gel conditions at both timepoints. The +gel niches show a similar down regulation of TCA cycle-associated metabolites as for hypoxic samples, but glycolysis associated metabolites are also downregulated. $N = 3$ for +gel 7 d, $n = 4$ for all other samples (d, e). **f** PerSCs were seeded for 7 days in the bioengineered models in the presence of $^{13}C_6$-glucose for 72 h, LC-MS was used to measure the conversion and abundance of $^{13}C_6$-labelled metabolites in the glycolysis pathway. Graphs show a fold change relative to $^{13}C_6$-labelled metabolites in PerSCs cultured in the −gel model, ±SEM, $n = 4$ material replicates. All statistics by one-way ANOVA with Bonferroni's multiple comparisons test, **a** comparisons to glass control, $*p < 0.05$, $**p < 0.001$, $****p < 0.0001$. Arb. u. arbitrary units. Source data are provided as a Source Data file.

niches, where PerSCs express high levels of nestin, cell death in response to oxidative stress was low (Fig. 6d). Reducing nestin by siRNA knock-down resulted in an increased sensitivity to oxidative stress, with a significant increase in PI-positive cells observed for cells in +gel niches (Fig. 6e, Supplementary Fig. 10c, d; note that all conditions are with $H_2O_2$). These results align with reports on the effects of oxidative stress on neuronal progenitors, where nestin-positive cells were more resistant to oxidative stress driven by $H_2O_2$-induced apoptosis[57].

### Engineered niches maintain LT-HSCs ex vivo

Next, we investigated whether our bioengineered niches could maintain LT-HSCs. Typically, HSC expansion media contains a cocktail of GFs. Here, we used a defined basic media (0% HSC Media; see Methods), which we supplemented with three necessary GFs (Flt3L (FMS like tyrosine kinase 3 ligand, 50 ng/mL); SCF (20 ng/mL); and TPO (25 ng/mL)) to make a standardised HSC media (called 100% HSC media). When HSCs were cultured in 100% media in standard tissue culture well plates, we refer to it as the 'gold standard' (GS) condition.

To test the niche conditions on maintenance of the CD34⁺ᵛᵉ/CD38⁻ᵛᵉ HSC population, PerSCs were first cultured in the bioengineered models for 14 days in stromal cell media in order to precondition the niches (see Methods). CD34⁺ᵛᵉ HSCs from human BM were then added to the niches when the media was changed to either 0% HSC media, or 100% HSC media. The cells were co-cultured for a further 5 days (Fig. 7a, Supplementary Fig. 12b), and then released with collagenase for flow cytometry analysis (Supplementary Figs. 11 and 12). Our strategy (as shown in Fig. 7b) excluded PerSCs from this analysis, by gating on CD45 positive cells and then defining and quantifying the CD34⁺ᵛᵉ/CD38⁻ᵛᵉ population. This population is commonly referred to as HSCs, but, in fact, contains both LT and ST-HSC populations. GS culture conditions showed a 6-fold increase in CD34⁺ᵛᵉ/CD38⁻ᵛᵉ cell numbers after 5 days of culture in the bioengineered models (Fig. 7c). When CD34⁺ᵛᵉ HSCs were co-cultured with PerSCs in −gel models, they expanded to a similar extent in both 0% and 100% HSC media, indicating that the PerSCs could support HSC expansion (Fig. 7c). However, when CD34⁺ᵛᵉ HSCs were co-cultured with PerSCs in hypoxic models or in the +gel niches, no expansion was observed in the CD34⁺ᵛᵉ/CD38⁻ᵛᵉ population (Fig. 7c).

As previously noted, LT-HSCs are inherently quiescent, making the expansion of this population, without differentiation, difficult[4]. Thus, strategies that robustly maintain LT-HSC functional properties in vitro are still unresolved. We, therefore, investigated if our bioengineered niches could maintain LT-HSC numbers rather than just a CD34⁺ᵛᵉ/CD38⁻ᵛᵉ heterogenous population. To enumerate LT-HSCs, LTC-IC assays were performed to identify the number of naive LT-HSCs within a CD34⁺ᵛᵉ mixed population. In this assay, defined culture conditions (feeder layers and media) are used in which ST-HSCs and progenitors die over 6 weeks. The potency of the remaining LT-HSCs was then tested using a colony-forming unit assay (CFUs), in which the

differentiation potential of individual LT-HSCs was tested in methylcellulose cultures for a further 7 days.

BM CD34⁺ᵛᵉ cells were co-cultured in engineered niches, and subsequently FACS sorted for CD45⁺ᵛᵉ/Lin⁻ᵛᵉ/CD34⁺ᵛᵉ markers to remove committed progenitors and PerSCs (Supplementary Fig. 11a). In the −gel condition with 0% or 100% HSC media, increased numbers of CD34⁺ᵛᵉ/CD38⁻ᵛᵉ HSCs were observed compared to the starting population (i.e., expansion) (Fig. 7c); however, fewer LT-HSCs were preserved compared to the gold-standard conditions (Fig. 7d). In the +gel niches with 100% HSC media, no CD34⁺ᵛᵉ/CD38⁻ᵛᵉ expansion was noted (Fig. 7c) and LT-HSC numbers were reduced demonstrating phenotypic drift (Fig. 7d). Importantly, the +gel niche with 0% HSC media maintained both total CD34⁺ᵛᵉ/CD38⁻ᵛᵉ cell numbers (Fig. 7c) and maintained a significantly higher number of LT-HSCs, similar to the expanded GS population (Fig. 7d). This indicates that the CD34⁺ᵛᵉ/CD38⁻ᵛᵉ expansion observed in the GS condition must have been in the ST-HSC/progenitor population as it did not lead to an increased number of LT-HSCs compared to +gel 0% (Fig. 7c, d). It is noteworthy that the subsequent differentiation profile of LT-HSCs from the +gel niche with 0% HSC media mimics that of the GS cultures, with no loss of lineage commitment, as indicated by the ability to reconstitute all blood cell progenitors (Fig. 7e, Supplementary Fig. 11b).

To confirm this data, we first used extended flow cytometry analysis to identify CD45⁺ᵛᵉ/CD41a⁻ᵛᵉ/CD16⁻ᵛᵉ/CD7⁻ᵛᵉ/CD38⁻ᵛᵉ/CD34⁺ᵛᵉ/CD90⁺ᵛᵉ/CD45RA⁻ᵛᵉ (LT-HSCs) from CD45⁺ᵛᵉ/CD41a⁻ᵛᵉ/CD16⁻ᵛᵉ/CD7⁻ᵛᵉ/CD38⁻ᵛᵉ/CD34⁺ᵛᵉ/CD90⁻ᵛᵉ/CD45RA⁺ᵛᵉ (ST-HSCs) (Supplementary Fig. 12a)[60]. This confirmed the 0% +gel niche retained LT-HSCs without expansion in the ST-HSC compartment observed in other conditions. Secondly, an in vivo engraftment study showed that CD34⁺ᵛᵉ cells recovered from the niche after culture had engraftment potential similar to that of freshly thawed cells, demonstrating retention of the LT-HSC population (Fig. 7f, Supplementary Fig. 13; we note one animal receiving PEA +gel cells failed to engraft). It is notable that CD34⁺ᵛᵉ cells after culture in GS media had significantly reduced in vivo engraftment potential (Fig. 7f), alongside an increase in the proportion of human MPPs to LT-HSCs in the BM (Supplementary Fig. 13c). Together, this further demonstrates growth accompanied by differentiation in this control condition (Fig. 7f, Supplementary Fig. 13).

PerSC phenotype was next considered with respect to nestin and nuclear HIF1α post-culture with CD34⁺ᵛᵉ cells, with and without GS media. With the use of GS (100%) media in −gel and +gel conditions, only low levels of nestin and HIF1α were observed (Supplementary Fig. 14a, b) in PerSC phenotype. With the use of 0% media, while low levels of nestin and HIF1α were seen in −gel cultures, significant increases in both were noted in +gel cultures (Supplementary Fig. 14a, b). This data indicates that the PerSCs maintain their niche phenotype best in the condition where the best LT-HSC phenotype is maintained (+gel, 0%).

To demonstrate the therapeutic potential of our engineered niches, we also investigated their ability to support CRISPR-edited

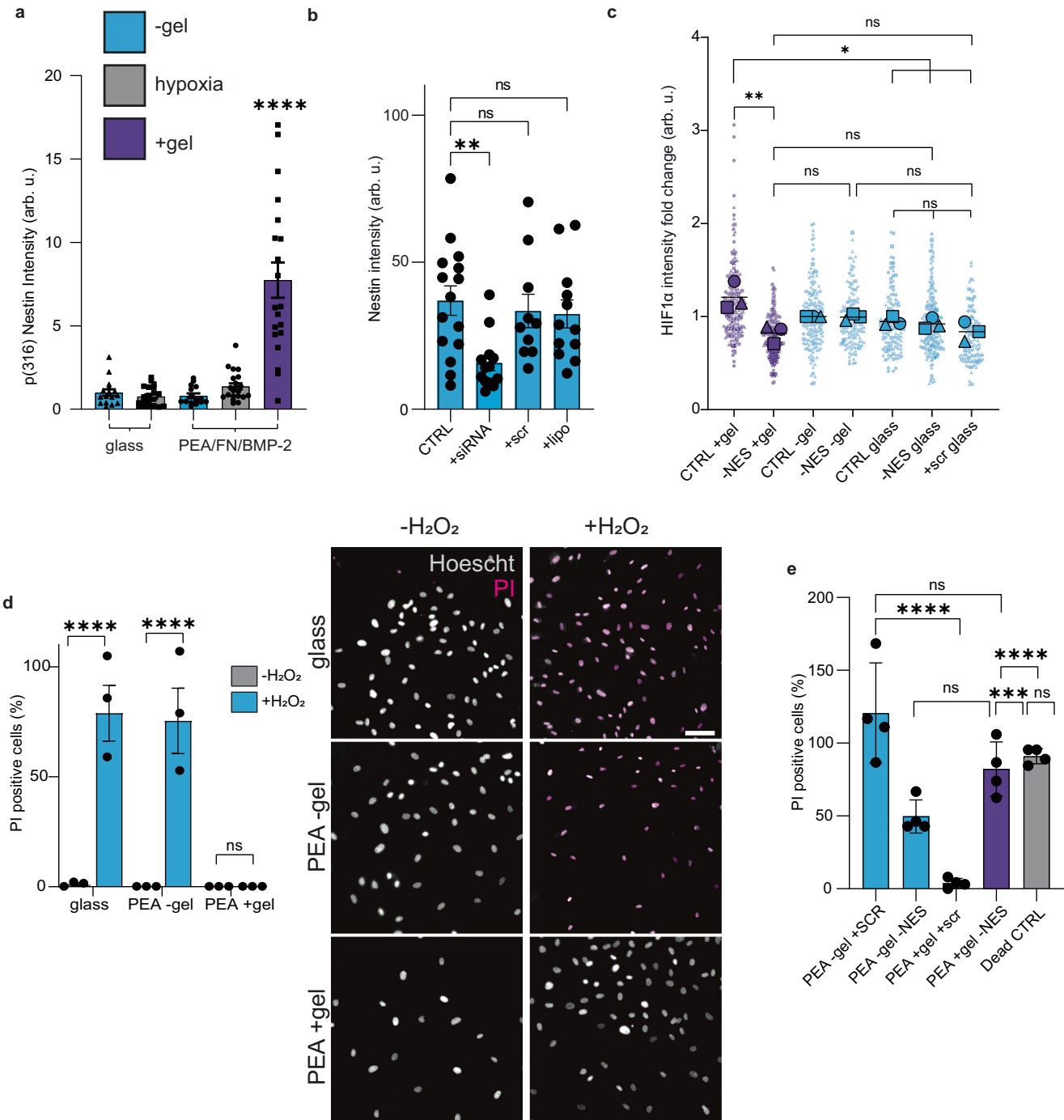

**Fig. 6 | Nestin is cytoprotective. a** Immunofluorescent detection of nestin phosphorylation at Th(316) indicates it is not incorporated into intermediate filament networks, but is instead soluble. Graph is mean ± SEM, $n = 4$ material replicates, where each point represents 1× image field normalised to cell number; −gel = blue, +gel = purple, hypoxia = grey. **b** siRNA knock-down of nestin in PerSCs was confirmed using immunofluorescence, graph shows mean ± SEM, $n = 3$ material replicates, where each point represents 1× image field normalised to cell number. **c** siRNA knock-down of nestin leads to the loss of activated HIF1α co-localisation to PerSC nuclei in +gel niches; no effect on HIF1α localisation was observed in hypoxia or −gel models (where PerSCs were nestin[low/negative]). $N = 3$, from 4 material replicates from 3 independent experiments with different donor cells. Scr = scrambled siRNA (control), lipo = lipofectamine only (control). **d** Nestin protects from oxidative stress. PerSCs in the bioengineered models were treated with 1 mM $H_2O_2$ to mimic

oxidative stress. $N = 3$ material replicates, graph shows mean ± SEM, grey = −$H_2O_2$, blue = +$H_2O_2$; representative images shown, scale bar is 100 μm, grey = Hoescht stained cells (live & dead); magenta = PI stained cells (dead). **e** siRNA knock-down of nestin in PerSCs and susceptibility to oxidative stress was assessed. In the +gel niches, nestin knockdown led to similar levels of cell death as in the −gel niches (nestin[low/negative] systems). All conditions were +$H_2O_2$, dead CTRL + 10 mM $H_2O_2$. $N = 4$ material replicates, graph shows mean ± SEM. **d, e** measured by the % of propidium iodide (PI) positive cells vs total Hoechst stained nuclei. Statistics by one-way ANOVA with Bonferroni's multiple comparisons test, **b** comparisons to untreated CTRL, *$p < 0.05$, **$p < 0.005$, ***$p < 0.0005$, ****$p < 0.0001$, ns non-significant. Arb. u. arbitrary units. Representative images corresponding to **a–c, e** in Supplementary Figs. 9, 10. Source data are provided as a Source Data file.

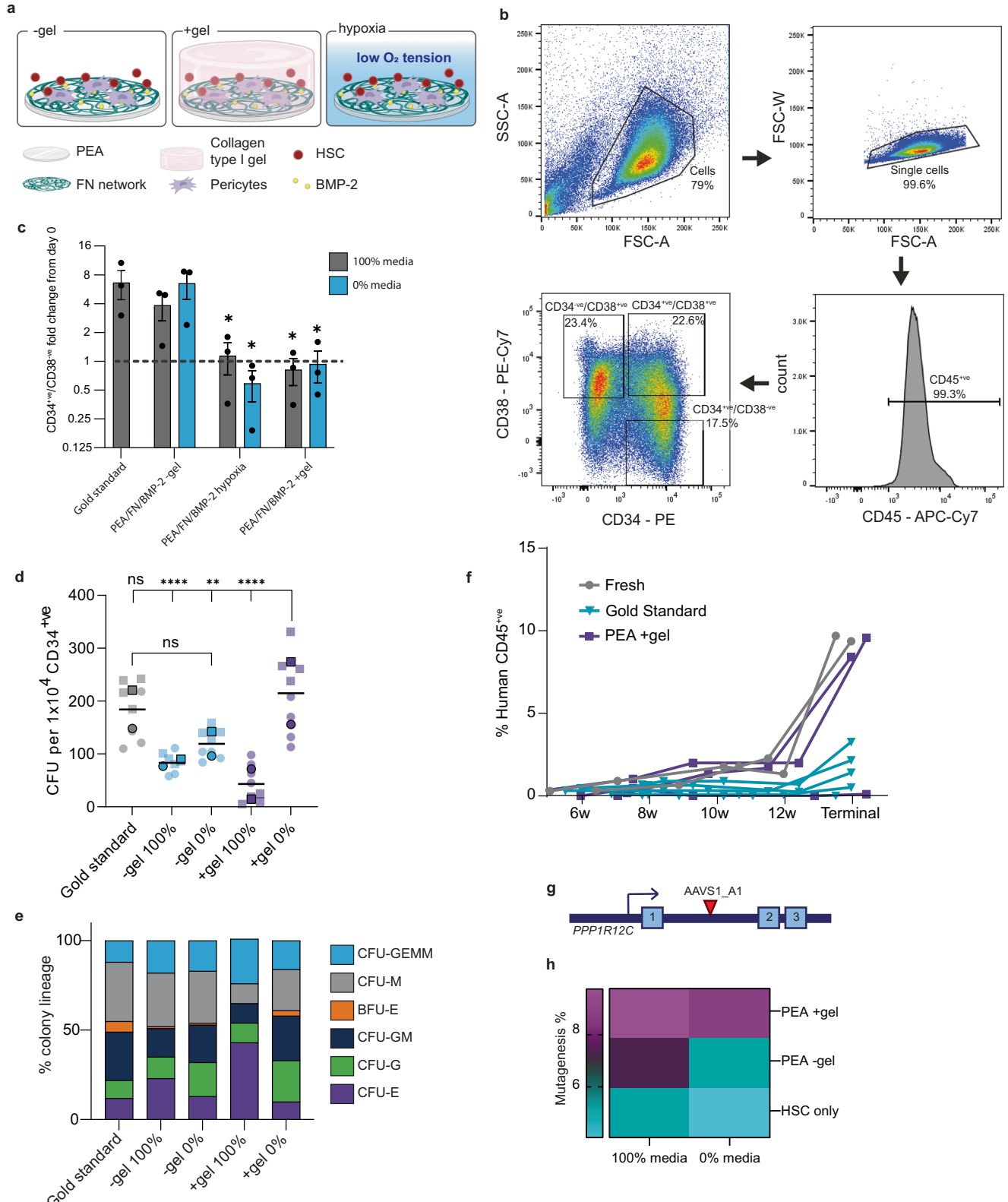

HSCs. The cellular repair of CRISPR-mediated double-strand DNA breaks in dividing cells primarily occurs via canonical non-homologous end-joining (NHEJ) or alternative end-joining processes like microhomology-mediated end-joining (MMEJ). It has previously been shown that quiescent LT-HSCs can be CRISPR edited primarily via NHEJ, while retaining engraftment potential[61]. As a proof of concept, we targeted the neutral AAVS1 (*PPP1R12C* intron 1) locus (Fig. 7g),

which can be edited in HSCs without loss of phenotype[61]. Using a NHEJ CRISPR strategy, we transfected primary human CD34[+ve] BM cells with ribonuclear protein (RNP) complexes of Cas9 and AAVS1 A1 single guide RNA (sgRNA) and seeded the cells into our bioengineered niches immediately after transfection. We then assessed if the different niche microenvironments could improve the survival of CRISPR-edited cells by assessing the fractions of edited HSCs after standard culture, or in

**Fig. 7 | Engineered niches support LT-HSC maintenance, in vivo reconstitution and CRISPR editing. a** Experimental set up schematic. **b** Representative gating strategy for HSC identification. Forward (FSC-A) versus side scatter area (SSC-A) plots identify viable cells; FSC-A versus FSC width (FSC-W) plots identify single cells. Gated on CD45[+ve] cells to exclude PerSCs, 3 gates used to identify CD34[+ve]/CD38[−ve] (LT- and ST-HSCs), CD34[+ve]/CD38[+ve] (stem and progenitor) and CD34[−ve]/CD38[+ve] (committed progenitors) populations. **c** Fold change in CD34[+ve]/CD38[−ve] cell number after 5 days co-culture in 100% (grey) or 0% (blue) HSC media relative to the number at day 0. $N = 3$, where each data point = 1 donor cell line from 3 independent experiments. Graph shows mean ± SEM. **d** Total number of colonies maintained in niche models. After 5 days co-culture, sorted CD34[+ve] cells were added to LTC-IC assays (6 weeks), resulting cells assessed using CFU differential assay. Total colony numbers were counted and represented as CFUs per $1 \times 10^4$ CD34[+ve] cells. $n = 2$ biological donors with 4 technical replicates per donor (donor 1 = circles, donor 2 = squares). **e** Lineage specification state of CFUs were identified; for colony definitions see Supplementary Fig. 11. **f** HSCs were cultured for 5 days in gold standard or PEA +gel niches, or freshly thawed (fresh) immediately prior to sorting of CD45[+ve]/Lin[−ve]/CD34[+ve] cells and injected in irradiated mice. Blood of recipient mice was analysed at 6, 8, 10, 12 and 14 weeks (terminal bleed). Graph shows percentage of human donor-derived CD45[+ve] cells. Number of recipient mice; fresh (positive control) = 2, grey; PEA +gel = 3, purple; gold standard = 5, blue. See also Supplementary Fig. 13. **g** AAVS1_A1 CRISPR target site at PPP1R12C gene locus. **h** Increased retention of AAVS1 CRISPR edited HSCs in +gel niches with 100% or 0% media, relative to gold standard or −gel conditions. Heatmap shows % AAVS1 mutagenesis, $n = 2$ biological donors, from 3 pooled material replicates. Statistics by one-way ANOVA with Bonferroni's multiple comparisons (**d**), or to Gold Standard condition (**c**), $*p < 0.05$, $**p < 0.005$, $***p < 0.0005$, $****p < 0.0001$. Source data are provided as a Source Data file. a, created with BioRender.com released under a Creative Commons Attribution-NonCommercial-NoDerivs 4.0 International license (https://creativecommons.org/licenses/by-nc-nd/4.0/deed.en).

the −gel and +gel niches, with either 100% or 0% HSC media. As expected, mutagenesis levels were low in these primary human HSCs. We achieved just ~5-6% mutagenesis in 100% HSC media (~2% in 0% HSC media) in both standard and −gel conditions (Fig. 7h). However, notably higher levels of CRISPR edited HSCs were retained in the +gel niche, in both 100% and 0% HSC media conditions at ~10% and ~9% respectively (Fig. 7h, Supplementary Fig. 15).

## Discussion

In this study, we set out to show that a combination of materials could be used to organize the ECM, GF presentation, and elastic environment of a putative BM niche. In vivo, the BM endosteum is densely perfused with a network of thin-walled arterioles[13,24]. These arterioles are closely associated with nestin[+ve] PerSCs[12,13,18], and HSCs isolated from this area show greater engraftment potential than their central marrow (sinusoidal) counterparts[62]. Here, we report an initial attempt to bioengineer this endosteal/arteriolar region of the BM niche, by promoting a niche-like phenotype in PerSCs that can support and maintain LT-HSCs, the HSC population that resembles those isolated from this niche region in vivo.

The +gel niche model developed here supports the expression of nestin in PerSCs, a phenotype that has been well-reported to be critical for the stromal-HSC supportive phenotype in vivo[12,13,18,27] and has been shown to have increased expression in stromal cells in soft environments[19]. Indeed, expression of nestin by PerSCs correlates with increased expression of other important niche factors, such as HSC maintaining cytokines, decreased oxidative phosphorylation/respiration[38,63], and increased cytoprotection[55–57]. We demonstrate that nestin-linked changes in PerSC respiration were central to the LT-HSC supportive phenotype. Nestin was further implicated in the persistent expression of nuclear HIF1α in the absence of low oxygen tension and without changes in the expression of hypoxia-signaling targets. This has parallels to cancer cells using HIF metabolism without hypoxia in the Warburg effect[54].

This approach towards LT-HSC maintenance is important, as most studies focus on expansion of the total heterogenous HSC population in vitro, and do not separate naïve LT-HSCs from other committed progenitor populations[10,28,30]. Currently, the expansion of LT-HSCs and the mechanisms required remain elusive. It is likely that clinically relevant expansions (e.g., 1 donor to multiple recipients) will require genetic manipulations/perturbation as naturally quiescent LT-HSCs will be required to enter the cell cycle and proliferate, then return to the resting state[64]. Until these fundamental principles of LT-HSC dynamics are understood and can be harnessed, systems that focus on their effective maintenance are required to help understand these dynamics and to improve therapeutic applications such as drug development, alloSCT, or gene editing.

Here, our data suggests that the use of common exogenous GF supplementation in combination with these niche co-cultures provides excessive stimulation to the HSC population and results in the loss of quiescence. However, by manipulating microenvironments to tune stromal cell physiology, LT-HSCs can be maintained in vitro without any exogenous supplementation. The CFU and in vivo data (Fig. 7; Supplementary Figs. 11 and 13) indicate that the CD34[+ve]/CD38[−ve] expansion (Fig. 7c) observed in the GS condition must have been in the ST-HSC/progenitor population, as it did not lead to an increase in CFU number (i.e., LT-HSCs) or to long-term engraftment in vivo, in fact, CFUs decreased, and observed engraftment was poor, compared to culture in the +gel niche. As such, we suggest that the use of the optimised +gel niche avoids the unwanted ST-HSC/progenitor expansion that comes with the concomitant sacrifice of LT-HSC numbers.

Furthermore, once we had established an LT-HSC maintenance niche and investigated a role for nestin in PerSC support phenotype, we went on to investigate a potential use for the niches in supporting CRISPR editing. Using an NHEJ-based CRISPR approach that facilitates the targeting of quiescent cells, such as LT-HSCs[61], we demonstrated that our bioengineered niches improve the survival of CRISPR-edited HSCs. This is important because of the very low abundance of LT-HSCs and shows that our niches can maintain these cells. For such approaches to be used successfully to gene edit HSCs for therapeutic purposes, CRISPR editing rates in LT-HSCs, and LT-HSC survival, will need to be as high as possible to ensure that enough of the edited population can engraft and rebuild the blood system. Recent work shows that stromal support for gene edited-HSCs (Dexter-like cultures) provides required HSC support factors and can provide better engraftment[65]. Bioengineered niches can be envisaged to offer further enhanced control of the gene edited-LT-HSC phenotype.

Our study has limitations in that it uses a natural hydrogel, collagen. The use of a synthetic hydrogel would have advantages in increased reproducibility/control of viscoelasticity, degradability, and biological milieu[66]. Further, while we advocate an approach of LT-HSC maintenance, the ability to robustly expand this population is deeply alluring. In the BM niche, the LT-HSCs are produced or, perhaps, dedifferentiated from ST-HSCs/progenitor cells[64]. Looking to bioengineer the sinusoidal niche, with different ECM constituents[39,40], different stromal support[13] and different oxygen tension[24], could thus be interesting and rewarding.

Such advances in bioengineering will need to progress in hand with an evolving understanding of LT-HSC quiescence and activation. For example, LT-HSCs, themselves, are heterogeneous and this could have practical importance in targeting specific populations of LT-HSCs with materials. CD112[low] LT-HSCs, are differentially quiescent from CD122[hi] LT-HSCs and have better-preserved self-renewal and regeneration capacity under regenerative stress[67]. Advancing biochemical information can also provide target pathways for bioengineers to

design materials to interact with, as we do with integrins and GFs here. For example, transcription factor EB (TFEB) has been shown to provide lysosomal regulation to drive digestion of membrane receptors, such as transferrin and insulin receptors, in order to limit LT-HSC metabolism and prevent differentiation[68]. The subcellular localisation and activity of TFEB are regulated by mechanistic target of rapamycin (mTOR)-mediated phosphorylation, which occurs at the lysosomal surface[69]. Materials platforms have also been implicated in changes in chromatin organisation[70–73] and, for LT-HSCs, understanding chromatin accessibility to suppress CCCTC-binding factor (CTCF), restrains LT-HSCs from transitioning to activated ST-HSCs[74]. These biological insights guide us for future bioengineered niche developments.

# Methods

## Ethical statement
This study uses PerSCs isolated from the adipose tissue of healthy consenting patients undergoing cosmetic lipectomy procedures or from patients undergoing breast reconstruction procedures using deep inferior epigastric perforators (DIEP) with prior written consent; as such, all donors from this procedure were female. Ethical approval for the collection of adipose tissue and subsequent research was granted by the South-East Scotland Research Ethics Committee 3 (SESREC03, reference no. 10/S1103/ 45). Human CD34+ve cells were either purchased from CalTag Medsystems or STEMCELL Technologies, or isolated from the bone marrow aspirates of patients undergoing joint replacement surgery with prior written consent. The permission to use the residual tissues was given by the Greater Glasgow and Clyde NHS Biorepository. CD34+ve cells used in this study are from patients aged 20-80 years old and are a mixture of male and female donors. 50/50 male/female CD34+ve cells were purchased. Both male and female cells were used in all experiments. In line with ethics and patient confidentiality, details of age and sex were blinded to the researchers carrying out tissue isolations and subsequent work. Due to limited donor availability via both procedures, no specific age group or sex was used in this study. All NRG-3GS mice were 8 -10 wk old males and were housed at the Beatson Research Unit (University of Glasgow, UK). Experimental protocols for working with animals were approved by the local AWERB committee and the national Home Office (PD6C67A47).

## Preparation of materials and ECM interfaces
Polymer sheets were obtained by radical polymerisation of a solution of either MA (methyl acrylate) or EA (ethyl acrylate) (Sigma-Aldrich, UK), using 1 and 0.35 weight percent benzoin (98% pure; Scharlau) as the photoinitiator. Polymerisation carried out up to limiting conversion. After polymerisation, low molecular mass substances extracted by drying under vacuum to constant weight. PMA and PEA were dissolved in toluene at a concentration of 6% or 9% and 2.5% or 12% dependent on batch. Spin casting was performed on glass coverslips at 2000 rpm for 30 s. Samples dried under vacuum at 60 °C for 2 h, and sterilised under UV light for 30 min before use[35]. FN from human plasma (Sigma-Aldrich, F2006) was adsorbed from solutions of 20 µg/ml for 1 h a room temperature (RT) and then washed thrice with phosphate-buffered saline (PBS). For GF adsorption, BMP-2 (50 ng/ml; Sigma-Aldrich, H4791) in PBS was used for 2 h at RT. Finally, samples were rinsed in PBS to eliminate non-adsorbed protein.

## Atomic force microscopy
FN was prepared in PBS (20 µg/ml), and a 200 µl droplet placed on surface of PEA and PMA-coated coverslips. The protein was adsorbed for 10 min, and remaining liquid removed from the surface. Surfaces were then washed twice in PBS, once with deionized, water and then dried under a stream of nitrogen before imaging. A JPK Nanowizard 4 (JPK Instruments) was used for imaging in tapping mode, using antimony-doped Si cantilevers with a nominal resonant frequency of 75 kHz (MPP-21 220, Bruker). The phase signal was set to 0 at a frequency 5–10% lower than the resonant frequency. Height and phase images were acquired from each scan, and the JPK data processing software version 5 was used for image analysis.

## FN adsorption assays
To quantify amount of FN adsorbed onto PEA or PMA surfaces substrates were coated with 20 µg/ml FN/PBS solution for 1 h, aspirate was collected, and FN quantified using the Pierce™ BCA Protein Assay Kit (ThermoFisher, UK). Quantitative immunofluorescence assays were carried out using the LI-COR in-cell western™ platform. FN (20 µg/ml) was adsorbed as previously described, samples were washed with PBS, blocked with 1% milk protein in PBS, and incubated with primary antibodies for total FN (polyclonal rabbit, Sigma Aldrich), HFN7.1 (monoclonal mouse, Developmental Studies Hybridoma Bank, USA), and P5F3 (monoclonal mouse, Santa Cruz Biotechnology, sc-18827) for 2 h. Substrates were washed 5× 0.5% Tween20 in PBS (PBST), followed by incubation o/n at 4 °C with LI-COR secondary antibodies (IRDye 800CW/700CW anti-rabbit/mouse secondary antibody, LI-COR, UK). Samples were washed 5× with PBST, followed by a final wash in PBS and dried before imaging on LI-COR Sa Odyssey scanner.

## BMP-2 adsorption
For quantification of BMP-2 adsorption, (50 ng/ml) BMP-2 solution was added to FN-coated polymers and after 2 h incubation solution was aspirated and collected in Protein LoBind Tubes (Eppendorf™). Enzyme-linked immunosorbent assays (ELISA) were then carried out as per manufacturer's instructions (R&D Systems, BMP-2 DuoSet ELISA kit, DY355). Briefly, ELISA plates were coated with capture antibody overnight, then blocked for 1 h with BSA. Standards, original solution, original solution at 20× dilution, and sample aspirates were then added to the plate, and bound BMP-2 was detected with biotinylated anti-human BMP-2. Streptavidin-HRP was added to plates for 20 min in the dark, followed by substrate solution (tetramethylbenzidine and peroxide) for 20 min, the reaction was then stopped by adding stop solution. Absorbance measured at 450 nm with wavelength correction at 570 nm.

## PerSC isolation
PerSCs were isolated from the adipose tissue of healthy adult donors undergoing cosmetic lipectomy procedures with prior written consent, or from patients undergoing breast reconstruction procedure, using DIEP, from adult donors. Incisions were made in the adipose tissue using a scalpel to divide the Scarpa's fascial layer, then tissue was mechanically disrupted, combined with PBS and centrifuged at 445 × g for 10 mins at RT, causing phase separation. The 3 phases include the top phase (liquid fat/oil), central phase containing the tissue of interest (adipose tissue), and the bottom phase (blood/fluid). Middle layer was removed and mixed with equal volumes of PBS/2% (v/v) fetal bovine serum (FBS) and centrifuged at 445 × g for 10 mins. Supernatant was discarded and the remaining stromal vascular fraction (SVF) pellet, was enzymatically digested with type II-S collagenase 1 mg/ml in DMEM/0.5% (v/v) BSA for 45 mins at 37 °C. Samples were centrifuged and supernatant discarded (containing oil and adipocytes) and pellets resuspended and strained through 400 µm, 100 µm and 70 µm cell pluristrainers to remove undigested material. Red blood cell lysis buffer was added to eliminate erythroid cells and SVF pellet resuspended in Fluorescent Activated Cell Sorting (FACS) buffer. Cells sorted on CD146+ve/CD45−ve/CD34−ve/CD31−ve phenotype (anti-CD146 clone P1H12, BV711, 563186; anti-CD45 clone H130, V450, 560367; anti-CD34 clone 581, FITC, 555821; anti-CD31 clone WM59, PE-Cy7, 563651. All BD Biosciences), using BD FACSAria cell sorter (BD Biosciences). Immediately after sorting, cells were then cultured as previously described[16]. Briefly, cells were cultured in endothelial growth medium

(EGM-2, Lonza) on 0.1% gelatin-coated plates in a humidified incubator with 5% $CO_2$ at 37 °C until passage 1.

## PerSC cell culture

PerSCs, after passage 1, were cultured in Dulbecco's Modified Eagle's Medium (DMEM, Sigma-Aldrich) with 20% FBS (Thermo Fisher), 1% nonessential amino acids (Sigma-Aldrich), 1% 100 mM sodium pyruvate (Sigma-Aldrich) and an antibiotic mix consisting of 10 mg/ml penicillin/streptavidin (Sigma-Aldrich), 200 nM L-glutamate (Sigma-Aldrich), and 0.5% Fungizone (Thermo Fisher). Cells were incubated in a 5% humidified $CO_2$ atmosphere at 37 °C. For cell culture on polymer substrates, cells were seeded at $1.5 \times 10^3$/cm² on 13 mm coverslips in 24 well-plates and cultured in DMEM containing 2% with human AB serum (HS; Sigma Aldrich, H4522) in a 5% humidified $CO_2$ atmosphere at 37 °C. Hypoxic culture carried out in hypoxic workstation (Ruskinn) at an oxygen tension of 1%, $CO_2$ 5% and 37 °C. Medium was exchanged every 3 days.

## CD34$^{+ve}$ cell isolation

Human CD34$^{+ve}$ cells were isolated from the bone marrow aspirates of patients undergoing joint replacement surgery. For isolation of CD34$^{+ve}$ cells, mononuclear cells were resuspended with FACS buffer and incubated with CD34$^{+ve}$ microbeads (Militenyi Biotec, #130-046-702) for 30 mins at 4 °C. Cell suspensions were transferred to LS Columns (Militenyl Biotec, #130-042-401) and passed through the magnetic Quadromacs Separation Unit (Militenyi Biotec, #130-090-976). Magnetically labelled CD34$^{+ve}$ cells were then collected, counted and cryopreserved.

## CD34$^{+ve}$ cell culture

CD34$^{+ve}$ cells from human BM were purchased from CalTag Medystems (experiments related to Fig. 7b–e) and STEMCELL Technologies or isolated from BM samples as above (Fig. 7f). CD34$^{+ve}$ cells were cultured in serum-free medium; Iscove's Modified Dulbecco Medium (IMDM) containing 20% BIT 9500 serum substitute (STEMCELL Technologies), 1% 200 mM L-glutamine (Sigma-Aldrich), 1% 10 mg/ml penicillin/streptomycin (Invitrogen) supplemented with Flt3L (50 ng/ml), SCF (20 ng/ml) and TPO (25 ng/ml) (all recombinant human, Peprotech, 300-19, 300-07, 300-18) (100% medium). CD34$^{+ve}$ cells were thawed, counted, and rested overnight in 100% medium at a density of $0.5 \times 10^6$/ml, then recounted, suspended in 100% or 0% media, and seeded into niche models at a density of $5 \times 10^4$ cells/ml, with 0.5 ml added to each well of 24 well plate. For conditions containing gels, the HSCs were seeded on top of the gel. Control (100% media only) was seeded into 24 well plates. Seeded cells were then cultured for 5 days. At least $1 \times 10^3$ cells were transferred to an Eppendorf tube and phenotyped at day 0 using flow cytometry.

## Collagen gel preparation and rheological measurement

Collagen type I gels were added to relevant (+gel) conditions after either 24 h (for short-term culture) or 72 h (for long-term culture). Rat tail collagen type I solution in 0.6% acetic acid (First Link, UK) was combined with human AB serum (HS; Sigma-Aldrich, H4522), 10× DMEM, 2% HS DMEM, adjusted to pH 8.2 using 0.1 M NaOH. 1 ml of solution was then added to 24 well-plates containing PEA coated 13 mm coverslips. The resulting gels were -1 cm thick. Rheological measurements were carried out using an Anton Paar Physical MCR301 rheometer. A parallel plate geometry (25 mm diameter, sandblasted) and 1.0 mm gap were used to measure time sweeps. The dynamic modulus of the hydrogel was measured as a frequency function, with frequency sweeps carried out between 0.1–15 Hz to measure the dynamic shear of the modulus as a function of strain. Measurements were repeated 3 times on gels from 3 different batches. Storage moduli (G´) values were extracted from the accompanying Kinexus software. The Young's modulus was then determined by taking the G´ and

multiplying by 3, assuming a Poisson's ratio of 0.5 in accordance with Hooke's law.

## Immunocytochemistry

PerSCs were cultured in niches for times indicated and fixed using 10% formaldehyde for 15 mins. For SCF and CXCL12 analysis, cells were cultured with 5 μg/ml brefeldin A (SigmaAldrich, B6542), for the final 24 h of culture before fixation. Cells were then permeabilized with 0.5% Triton-X for 5 mins and blocked using 0.5% BSA/PBS for 2 h at RT. Primary antibodies were then added in blocking buffer: anti-nestin [10C2] 1:200 (mouse monoclonal, Abcam, ab22035); anti-p(Th316)-nestin [a-4] at 1:200 (mouse monoclonal, Santa Cruz Biotechnology, sc-377538); anti-HIF1α [EP1215Y] 1:300 (rabbit monoclonal, Abcam, ab51608); anti-vimentin [SP20] 1:300 (rabbit monoclonal, Thermo Fischer, MA5-16409); anti-CXCL12 1:200 (monoclonal mouse, R & D Systems, MAB350); anti-SCF 1:200 (rabbit polyclonal, Abcam, ab64677). Cells were then washed 5 × 5 mins with PBST and biotinylated secondary antibodies (1:50; Vector Laboratories) were added in blocking buffer for 2 h at RT. Cells were again washed 3 × 5 mins in PBST and incubated with fluorescein isothiocyanate-conjugated streptavidin (1:50; Vector Laboratories) in blocking buffer for 30 mins at RT. Nuclei were stained using VECTASHIELD mountant with 5',6-diamidino-2-phenylindole nuclear stain (DAPI; Vector Laboratories). Samples were then mounted onto glass slides and visualised using an Axiophot microscope or Evos Cell imaging system (Thermo Fischer) and analysed using ImageJ software (National Institute of Health). HIF1α nuclear co-localisation was analysed using a custom-developed pipeline on CellProfiler software (version 2.1.1). Images were background corrected and analysis normalised to cell number via DAPI/Hoescht staining, or individual cells measured, as stated in figure legends.

## Flow cytometry

To phenotype PerSCs and HSCs after culture, cells were harvested from niche systems using collagenase D (2.5 mg/mL in PBS; Sigma-Aldrich) and TripLE™ (Thermo Fisher). Cells were passed through a 70 μm filter, and stained on ice for 30 mins, using antibodies outlined in supplementary Tables 3 and 4 for PerSC and HSC markers, in flow cytometry buffer (PBS supplemented with 0.5% BSA and 0.5 mM EDTA). Cells were then washed twice with PBS and analysed using an BD FACSCanto ll (BD Biosciences). The BD FACSAria Cell sorter was used for CD34$^{+ve}$ sorting for LTC-IC and in vivo assays. The gating strategy is shown in Supplementary Figs. 5 and 13. Flow cytometry files were analysed using FlowJo software (version 10.5.3, FlowJo LLC, USA). Single-cell suspensions from in vivo experiments were prepared for phenotypic analysis by flow cytometry as described previously[75]. All antibodies for in vivo analysis are shown in supplementary Table 5. All samples were suspended in 100 μl FACS buffer prior to acquisition. Flow cytometry data were acquired using a FACSCantoII flow cytometer (BD Biosciences) using the FACS Diva software package and analysed using the FlowJo software package (Tree Star, Inc., Ashland, OR).

## RNA-seq

Sequencing libraries then prepared from total RNA using the Illumina TruSeq Stranded mRNA Sample Preparation Kit. Libraries were sequenced in 75 base, paired-end mode on the Illumina NextSeq 500 platform. Raw sequence reads were trimmed for contaminating sequence adaptors and poor-quality bases using the programme Cutadapt[76]. Bases with an average Phred score lower than 15 were trimmed. Reads that were trimmed to <54 bases were discarded. The quality of the reads was checked using the FastQC programme (http://www.bioinformatics.babraham.ac.uk/projects/fastqc/) before and after trimming. The reads were "pseudo aligned" to the transcriptome using the programme Kallisto[77]. The differential expression for the analysis groups was assessed using the Bioconductor package

DESeq2[78]. This was provided as a service by Glasgow Polyomics Facility. One hypoxic sample failed RNA quality control. Heatmaps in Fig. 3g, h were subsequently generated using Cluster 3.0 and Java Treeview 3.0 software, with average linkage clustering method. To assess expression of transcripts related to an immunomodulatory phenotype, the distribution of reads across genomic features was quantified using the R package Genomic Ranges from Bioconductor Version 3.065. Differentially expressed genes were identified using the R package edgeR from Bioconductor Version 3.066. For Fig. 4a–c and Supplementary Fig. 6, analyses were performed within the R statistical computing framework using packages from BioConductor. Feature counts were utilised to quantify reading counts. Human ENSEMBL gene ID to gene symbol conversion was performed in BioTools (https://www.biotools.fr). The DEseq2 BioConductor package was used for outlier detection, normalisation and differential gene expression analyses. Genes passing a threshold of Wald test $p$ values and adjusted $p$ value < 0.05 and a log2 fold change >1 were considered as differentially expressed. The names of differentially expressed genes along with their respective log2fold changes were inputted into PathfindR package. Gene Ontology (GO) enrichment was conducted with PathfindR. Only pathways with FDR ≤ 0.05 were considered as differentially enriched. The differentially expressed genes involved in every pathway were extracted and their $z$ scored fold changes in expression were presented as heatmaps.

## VEGF ELISA
PerSCs were cultures in niche systems and media supernatant was collected in Protein LoBind Tubes (Eppendorf™) at days 7 and 14 and stored at −20 °C. ELISA for VEGF was then carried out as per manufacturer's instructions (R&D Systems, VEGF Quantikine ELISA kit, DVE00). Standards and media aspirate were added to the ELISA plate in duplicate. Absorbance measured at 450 nm with wavelength correction at 570 nm.

## Hypoxyprobe™ assay
PerSCs were cultured in niche models for 6 days, medium was then changed to fresh medium containing 200 μM pimonidazole (Hypoxyprobe™, HP1-200kit), and cultured for a further 24 h. Cells were then fixed, permeabilized, and blocked as described for immunocytochemistry. Mouse monoclonal anti-pimonidazole (1:200; Hypoxyprobe™, HP1-200kit) was added in blocking buffer and incubated overnight at 4 °C. Cells were washed 3 × 5 mins in PBST and incubated with anti-mouse secondary (1:50, Texas Red; Vector Laboratories) for 2 h at RT. Cells were washed 3 × 5 mins in PBST and mounted onto glass slides with DAPI nuclei stain VECTASHIELD mountant (Vector Laboratories). Samples were visualised on Evos Cell imaging system (ThermoFisher) and analysed using ImageJ software (National Institute of Health).

## Metabolomics
Whole-cell metabolomic analysis was performed on cell lysates isolated from PerSCs cultured in niche systems for 7 or 14 days. Substrates were washed with ice-cold PBS, and cells lysed in extraction buffer (PBS/methanol/chloroform at 1:3:1 ratio) for 60 mins at 4 °C with constant agitation. Lysates were then transferred to cold Eppendorfs™ and spun at 13000 g at 4 °C for 5 mins to remove debris and stored at −80 °C. Cleared extracts were used for hydrophilic interaction LC/MS analysis (UltiMate 3000 RSLC, (ThermoFisher), with a 6 150 × 4.6 mm ZIC- pHILIC column running at 300 μl/min-1 and Orbitrap Exactive). A standard pipeline, consisting of XCMS (peak picking), MzMatch (filtering and grouping) and IDEOM (further filtering, post-processing and identification) was used to process the raw mass spectrometry data. Identified core metabolites were validated against a panel of unambiguous standards by mass and predicted retention time. Further, putative identifications were generated by mass and predicted

retention times. Heatmaps of selected metabolites and PCA plots were generated using MetaboAnalyst software (version 4.0). This was provided as a service by Glasgow Polyomics Facility.

## $^{13}C_6$-Glucose metabolomic tracing
PerSCs were seeded onto PEA/FN/BMP-2 niche systems and allowed to grow for 72 h. Cells were then washed and media was changed to basal media comprising 25% normal glucose and 75% $^{13}C_6$-Glucose (Cambridge Isotopes Ltd). Extraction performed in PBS/methanol/chloroform at 1:3:1 ratio, after 3 days incubation and prepared for LC-MS. The LC-MS platform consisted of an Accela 600 HPLC system combined with an Exactive (Orbitrap) mass spectrometer (ThermoFisher). Two complementary columns were used; the zwitterionic ZICpHILLIC column (150 mm × 4.6 mm; 3.5 μm, Merck) and the reversed phase ACE C18-AR column (150 mm × 4.6 mm; 3.5 μm Hichrom) and in both cases sample volume was 10 μl at a flow rate of 0.3 ml/min. Eluted samples were then analysed by mass spectrometry. Raw data from LC-MS of $^{13}C$-labelled extracts was processed to generate a combined PeakML file[79]. Further analysis using mzMatch-ISO in R[80] generated a PDF file containing chromatograms used to check peak-shape and retention time, and a tab-delineated file detailing peak height for each isopotologue, which was used to calculate percentage labelling. Samples normalised to cell number. Total $^{13}C_6$-Glucose incorporation was calculated by totalling incorporation excluding up to C2 to eliminate natural incorporation. This was provided as a service by Glasgow Polyomics Facility.

## siRNA NES silencing NES
PerSCs were seeded into niche models in antibiotic free 2% HS DMEM, after 24 h media was changed to Opti-MEM™ (ThermoFisher) and siRNA transfection was carried out 24 h later. Pre-designed Silencer™ Select siRNA NES (ID: 21141), or Silencer™ Select negative control number 1 siRNA (both Thermo Fisher), was incubated at RT for 20 mins with Lipofectamine™ RNAiMAX (ThermoFisher) in Opti-MEM™ and added to PerSCs in niches. 6 h later, Opti-MEM™ +2% HS was added to control and −gel conditions. For +gel, media containing siRNA was collected and used to prepare collagen gels as described above. Once collagen had set, Opti-MEM™ +2% HS was added. Cells were then cultured for 72 h and either fixed for immunofluorescence analysis or cytotoxicity assays carried out.

## Cytotoxicity assay
Cells were cultured in niche models for time points indicated. Collagen gels were removed and 1 mM $H_2O_2$ added for the final 6 h of culture. Hoechst (33342) nuclear stain (ThermoFisher, R37605) and PI (Biolegend, 421301) viability stain was then added and imaged on EVOS M7000 imaging system using the on-stage incubator.

## LTC-IC assay
LTC-IC assays were carried out on sorted CD45$^{+ve}$/Lineage$^{−ve}$/CD34$^{+ve}$ cells harvested after 5 days of culture in niche systems (supplementary Table 4). Engineered stromal fibroblast feeder layers were first established. M2-10B4 (overexpressing IL-3 and G-CSF) and Sl/Sl (overexpressing IL-3 and SCF) were passaged at ~90% confluence (cell lines gifted from StemCell Technologies). Cells were grown for 2 weeks prior to use in selection agents to select stromal cells expressing long-term cell maintenance factors (M2-10B4, 0.4 mg/ml G418 and 0.06 mg/ml Hygromycin B; Sl/Sl, 0.8 mg/ml G418 and 0.15 mg/ml Hygromycin B) G418 and Hygromycin B both StemCell Technologies (03812, 03812). Cells at ~80% confluence were irradiated with 8000 cGy, trypsinised. M2-10B4 and Sl/Sl were mixed at 1:1 ratio at a final concentration of 1.5 × 10⁶/ml. Cells were then seeded into collagen coated 24 well plates (Thermo Fisher, A1142802) for 24 h before adding sorted cells. Cultures were then maintained for 6 weeks in MyeloCult™ (StemCell Technologies, 05150) supplemented with 1 × 10$^{−6}$ M

hydrocortisone (StemCell Technologies, 74142), with half media exchanges twice per week.

## CFU assay

Cells were harvested and resuspended in MethoCult™ (StemCell Technologies, 04435) from LTC-IC assays after 6 weeks. Two replicates were seeded for each condition in 35 mm dishes and incubated at 37 °C in 5% $CO_2$ for 7 days. After 7 days total colonies were counted and phenotyped using a light microscope.

## Xeno-transplantations

NOD-RAG-γc$^{-/-}$ mice constitutively expressing human IL3, GMCSF and Steel factor (NRG-3GS[81]) were used as hosts (8–10 wk old; male) for transplantation, to establish the lineage potential of human HSCs derived from indicated culture conditions in vivo. Human BM CD34$^{+ve}$ cells were cultured in GS ($n = 5$ mice) or PEA +gel ($n = 3$ mice) niches as previously described and CD45$^{+ve}$/Lineage$^{-ve}$/CD34$^{+ve}$ cells harvested after 5 days of culture by FACS. Freshly thawed CD34$^+$-enriched HSCs isolated from BM (StemCell Technologies) were used as a positive control for engraftment ($n = 2$ mice). NRG-3GS mice were sub-lethally irradiated with two doses of 1 Gy, 3 h apart. The irradiated mice were transplanted within 24 hr of the last dose with human HSCs via tail vein injection (200,000 cells/mouse). After six weeks, blood sampling (~20 µl) was performed every 2 weeks to track the progress of human cell engraftment by flow cytometry (hCD45-APC-Cy7 vs mCD45-PerCP), preparing the blood samples for flow cytometry using Easy-Lyse (Agilent Technologies) following the manufacturer's instructions. After 12 wk, mice were sacrificed and BM, spleen and blood were collected for analyses. Spleens were weighed and cells extracted in FACS buffer (PBS/2% FBS). BM was recovered by crushing the ilia, femurs, and tibia of each mouse using a pestle and mortar in FACS buffer. The cells collected from the BM and spleen were filtered through 70 µm mesh and enriched for haemopoietic lineage cells by density centrifugation using Lympholyte-Mammal (VWR, Lutterworth, UK), centrifuging the cell suspension for 20 min at $625 \times g$ at RT. Thereafter, the cells were washed in FACS buffer, centrifuged at $500 \times g$ for 10 min at RT. Cells were counted using a haemocytometer and prepared for flow cytometry.

## CRISPR

CD34$^{+ve}$ cells were thawed 24 h before electroporation and cultured in IMDM + BIT + 100% cytokine media without antibiotics. Ribonucleoprotein (RNP) complexes were prepared by mixing Cas9 (IDT) at 12 µM and AAVS1_A1 sgRNA (GTCACCAATCCTGTCCCTAG, Synthego) at 12 µM, in electroporation buffer R (ThermoFisher) with 2% glycerol[61], and incubated for 15 min at RT immediately before electroporation. Electroporation was performed using the Neon™ transfection system, with 10 µl tips (ThermoFisher), as per manufacturers protocol. The optimal electroporation conditions for CD34$^{+ve}$ cells was 1700 V, 20 ms, one pulse. CD34$^{+ve}$ cells were resuspended in buffer R and mixed with assembled RNPs. Electroporated cells were resuspended in either 100% or 0% cytokine IMDM + BIT and 0.5 ml added into niche models at a density of $5 \times 10^4$ cells/ml. Cells were then cultured for 5 days, and harvested as previously described using collagenase D, with final resuspension into 30 µl of Quick Extract™ reagent (Lucigen). For mutagenesis analysis, Quick Extract samples were diluted 1:1 with fresh QuickExtract and heated to 65 °C for 6 mins, vortexed for 15 s, and heated to 98 °C for 2 mins. Genomic PCR was carried out using Herculase II Fusion DNA Polymerase (Agilent) (Forward primer: CCCCGTTCTCCTGTGGATTC, Reverse primer: ATCCTCTCTGGCTC-CATCGT). PCR products were run on 1.2 % UltraPure™ Agarose (Invitrogen) gels, purified (QIAquickPCR Purification Kit, Qiagen), and Sanger Sequenced (Eurofins Genomics). Mutagenesis analysis was carried out by TIDE analysis[82] (http://shinyapps.datacurators.nl/tide/) (Supplementary Fig. 5).

## Trilineage differentiation of PerSCs

PerSCs at ≤passage 5 were cultured in differentiation medias for 4 weeks: osteogenic media–DMEM with 10% FBS, 0.1 µM dexamethasone (Sigma, D2915), 100 µM Ascorbate-2-phosphate (Sigma, A8960), 10 mM β-Glycerophosphate disodium salt hydrate (Sigma, G9422); Adipogenic media–DMEM with 10% FBS, 1 µM dexamethasone, 500 µM 3-isobutyl-1-methylxanthine (Sigma, I5879), 1.8 µM Insulin (Sigma, I9278), 100 µM indomethacin (Sigma, I7378); Chondrogenic media–100 nM dexamethasone, 100 nM Ascorbate-2-phosphate, 1% (v/v) insulin, transferrin, selenium (Thermo Fischer, 41400-045), 40 mg/mL L-proline, 10 ng/mL TGFβ3 (Peprotech, 100-36E). Cells were then fixed using 10% formaldehyde for 15 mins at RT. After washing with PBS, osteogenic fixed cells were stained with 2% (w/v) Alizarin Red solution (pH 4.1 to pH 4.3) for 15 min at room temperature. After staining, cells were washed in deionized water. For adipogenic staining, fixed cells were washed with distilled water three times, rinsed with 60% (v/v) isopropanol. Oil Red O solution was then added to the cells, and cells were incubated at room temperature for 15 min. Dye solution was removed, and cells were washed again with 60% (v/v) isopropanol, washed three times in distilled water. For chondrogenic staining, cells were fixed with 0.1 % glutaraldehyde in PBS for 20 min at RT. Rinsed 3× PBS, rinsed 1× with 1 % acetic acid solution for 10–15 s and stained with 0.1 % Safranin O solution (Merck) for 5 min. Cells were then washed 3× PBS. Cells were then imaged on an inverted microscope (Olympus, PA, USA) operated through Surveyor software (v.9.0.1.4, Objective Imaging, Cambridge, UK). Images were processed using ImageJ [v.1.50 g, National Institutes of Health (NIH), USA].

## Lactate assay

PerSCs were cultured in niche systems in DMEM with 2% dialysed FBS (ThermoFisher, A3382001), cell media supernatant was aspirated at 7 and 14 days and stored at −20 °C. Lactate levels in supernatant were then measured using the Lactate-Glo™ assay (Promega, J5021) as per manufacturers protocol. Briefly, samples were diluted 1:100 and mixed 1:1 with Lactate Detection Reagent in 96-well plate and incubated for 1 h at RT. Luminescence was read using PHERAstar FSX microplate reader (BMG Labtech).

## In-cell western assay for LDH

PerSCs were cultured in niches for 7 days then fixed using 10% formaldehyde for 15 min. The cells were then permeabilized with 0.5% Triton-X for 5 mins and blocked using 0.5% milk protein in 1× PBS. Anti-lactate dehydrogenase (LDH) [EP1566Y] (rabbit monoclonal, Abcam, ab52488) was added at 1:200 in 0.5% milk PBS and incubated at 4 °C overnight, followed by 5 washes in PBST. CellTag 700 stain (LI-COR, 926-41090) was used as the reference control, and was added with LI-COR anti-rabbit secondary antibody (926-32211), at 1:2000 in 0.5% milk PBST and incubated at RT on a shaker for 2 h, followed by 5× 5 min washes with PBST. The quantitative spectroscopic analysis was carried out using the LI-COR Odyssey Sa.

## Statistics and reproducibility

Three or more material replicates were used for independent experiments with different biological donors, unless otherwise indicated in the figure legends. For CD34 + HSC cells used for in vivo analysis, 10 donors were pooled. Were multiple replicates pooled, this was on a per-application basis, where larger amounts of materials are required for certain techniques. No statistical method was used to predetermine the sample size. Shapiro–Wilks normality tests were performed, and statistical analysis used to calculate $P$ values was by one-way analysis of variance with Tukey's or Bonferroni's multiple comparison tests for normal and non-normal data, respectively. $T$ tests used were indicated in figure legends. All statistical analysis was performed using GraphPad Prism 10 software. Refer to the Source Data file for the details on statistical analysis. One sample was removed from metabolomics

analysis as it failed to produce readable peaks. One sample was removed from RNA-seq analysis due to insufficient RNA quality. Quality reports and peak intensity tables can be provided. The investigators were not blinded to allocation during experiments and outcome assessment; except for in vivo experiments, where the investigator was blinded upon the xenotransplantation and flow cytometry data collection. Microscopy is performed from randomised regions to avoid ROI bias.

## Reporting summary

Further information on research design is available in the Nature Portfolio Reporting Summary linked to this article.

## Data availability

The LC-MS data and all datasets supporting the findings in this study are openly available under the Creative Commons Attribution (CC-BY) license at: https://doi.org/10.5525/gla.researchdata.1326. Due to the large size of the data files, access must be requested. Access can be obtained from the corresponding authors. Requests will be processed within 2 working days. The RNAseq data generated from PerSCs in this study have been deposited in NCBI's Gene Expression Omnibus and are accessible through GEO Series accession number GSE265789. Source data are provided with this paper.

## Code availability

The custom CellProfiler pipeline used for nuclear HIF1α measurements is openly available in the DOI file at: https://doi.org/10.5525/gla.researchdata.1326. The data are licenced with a CC-BY open access licence and access requests to the corresponding authors are handled within two working days.

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

## Acknowledgements

We thank Dr. Gillian Higgins for the provision of adipose tissue to isolate PerSCs and Dr. Mark Sprott for help with ethyl acrylate synthesis. We thank Carol-Anne Smith and Vineetha Jayawarna for their technical assistance. We also thank the University of Glasgow Polyomics Facility, particularly Dr Karl Burgess and Dr Gavin Blackburn. This work was supported by BBSRC grant BB/N018419/1, EPSRC grant EP/P001114/1, MRC grant MR/R005567/1 (all M.J.D.) and the Carnegie Trust grant number RIG009892 (H.D.).

## Author contributions

H.D., E.R., H.W., A.M.M., A.G.W., M.S.S. and M.J.D., conceived and designed the experiments. H.D., E.R., Y.X., R.H., A.T., W.S.D.-B., J.C., P.M.T., A.C. performed the experimental work. N.J. and Y.X. carried out RNA-seq analysis. K.M.D. and J.H. carried out the animal work. D.M. provided the bone marrow for CD34$^{+ve}$ cell isolation. B.P. and C.W. provided PerSCs. H.D. and M.J.D. wrote the manuscript. H.W., A.M.M., A.G.W. and M.S.S. revised the manuscript. H.D. and M.J.D. secured the funding.

## Competing interests

The authors declare no competing interests.

## Additional information

¹Centre for the Cellular Microenvironment, School of Molecular Biosciences, The Advanced Research Centre, 11 Chapel Lane, University of Glasgow, Glasgow G11 6EW, United Kingdom. ²School of Cancer Sciences, Wolfson Wohl Cancer Research Centre, University of Glasgow, Glasgow G61 1QH, United Kingdom. ³School of Cancer Sciences, Paul O'Gorman Leukaemia Research Centre, Gartnavel General Hospital, University of Glasgow, Glasgow G12 0YN, United Kingdom. ⁴Centre for the Cellular Microenvironment, Division of Biomedical Engineering, James Watt School of Engineering, The Advanced Research Centre, 11 Chapel Lane, University of Glasgow, Glasgow G11 6EW, United Kingdom. ⁵Department of Trauma and Orthopaedics, Queen Elizabeth University Hospital, Glasgow G51 4TF, United Kingdom. ⁶Institute of Inflammation and Ageing, University of Birmingham, Queen Elizabeth Hospital, Birmingham B15 2WB, United Kingdom. ⁷MRC Centre for Regenerative Medicine, The University of Edinburgh, Edinburgh EH16 4UU, United Kingdom. ✉e-mail: manuel.salmeron-sanchez@glasgow.ac.uk; matthew.dalby@glasgow.ac.uk

