## [Peer Review File · Nature Communications]

REVIEWER COMMENTS

Reviewer #1 (Remarks to the Author):

Donnelly et al. have developed a bioengineered niche supporting LT-HSC quiescence. The niche is composed of ECM and perivascular stromal cells (PerSCs) on PEA and recapitulates the regions in close proximity to the endosteal surface. The unique microenvironments was able to keep PerSCs functionally and metabolically naïve and cytoprotective by enhancing nestin and HIF1 α expression. The naïve PerSCs produced factors that maintain LT-HSCs quiescence, avoiding exhaustion by proliferation and differentiation. These results show that the bioengineered system may be useful to maintain or expand fully-functional HSCs for a source of clinical HSC transplantation or genetic editing of HSCs for treating hereditary blood disorders. However, there are several issues that have to be addressed for publication in the nature communications.

Major concerns

A LT-HSC activity was evaluated only in vitro. Functional LT-HSC activity should be shown in vivo by xenotransplantation models.

The factors that make the bioengineered system “endosteal niches for LT-HSCs” should be investigated and identified. e.g. What factors distinguished “endosteal niches for quiescence” and “the ones for proliferation or differentiation”?

“The gold standard condition”, in which a cocktail of several cytokines were just add to media, seems to be as good as, even better than, the bioengineered niche not only for maintaining quiescence (LT-HSC numbers in LTC-IC assay), but also for allowing their expansion. The authors should clarify the advantages for using the bioengineered system compared to conventional culture systems.

Specific comments

In Fig1, the bioengineered niche on PEA showed several domains of FN exposed to bind integrins and trap growth factors. What growth factors for osteogenesis or niche functions were trapped in the structure?

In Fig2c,d, quality of the immunofluorescence images are not enough to show individual cells.

Images with higher magnification and resolution need to be shown.

In Fig2e, this point is related to Fig1, higher amounts of SCF was observed in the PEA+gel condition. Is that due to mRNA expression, protein synthesis, degradation or secreted one trapped by ECM?

How about other cytokines, cxcl12, SPP1, etc. essential for niche functions of PerSCs? What is the key factor that endows PerSCs with ability to support LT-HSCs?

In Fig2f, FACS analysis and the heatmap exhibit higher expression of the surface markers.

Is the population expressing the markers higher or Is the cellular numbers expressing them increased? Dot plots and histograms will be useful to show this point.

In Fig2h, although expression of “differentiation genes” are shown, functional assays such as in vitro tri-lineage differentiation have to be performed.

In Fig3, the core data that characterize the differences among the three culture conditions have to be confirmed by qPCR.

In Fig4a and Fig5d, the images have almost no information due to low quality and resolution.

In Fig6d,e, this point is raised as a major comment, please clarify the advantages for using the bioengineered system compared to conventional culture systems.

In Fig6g, CRISPR edited HSCs exhibited better survival in all of the three condition with 100% media. What is the biological differences between LT-HSCs and CRISPR edited HSCs?

Reviewer #2 (Remarks to the Author):

Using bioengineering approaches, Donnelly et al. present a model to recreate the biomechanical and extracellular matrix properties of the bone marrow (BM) endosteal niche in which perivascular stromal cells (PerSC) phenotype is more likely able to support long term-HSC (LT-HSC) maintenance. Hence, PerSC cells are grown on a Poly(ethyl acrylate) (PEA) polymer presenting fibronectin (FN) and BMP-2 overlaid by collagen type-I hydrogels with a Young modulus close to the one observed in the central BM (80Pa). In these conditions, PerSCs retain a naïve phenotype, including nestin expression. The authors showed that nestin expression by the PerSC play cytoprotective role involving HIF1a expression via metabolic regulations. This bioengineered stromal feeder niche maintains functional LT-HSC. The authors also give a proof a principle that this system allow HSC CRISPR/Cas9 gene editing. The acquisition and in vitro maintenance of functional haematopoietic stem cells (HSC) as primitive as possible for allogenic transplantation purposes is one of the current challenges in the field of haematopoiesis, therefore the presented work is very interesting and relevant to the field.

Overall the work is of high quality. Thinking about how best to compile and present their work so that all the conclusions are as strong as possible, I will list below a few comments aimed at either improving the clarity of the manuscript for the wide readership of the journal or at providing some further mechanistic insights to the observations presented.

The aim of Fig2 and 3 is to characterize PerSC cells phenotype on the +gel condition. From the images presented in Fig2.c it looks like the number of cells is increased compared to the other conditions. Even if the RNAseq data confirm the nestin overexpression in the +gel condition, the quantification of the images should take the number of cells in account. This could also be applied to Sup Fig1a.

Same comment regarding vimentin expression quantification in figure 1d.

It looks like the PerSC cells didn't migrate inside the overlaid collagen hydrogel (from fig1 a scheme and Fig 6b doesn't show a CD45- population). Is it the case? If yes, is it an expected result? If PerSC do migrate inside, what is the proportion of PerSC cells found in the gel and in contact/proximity with HSC once co-cultured?

It is unclear how the different time points were chosen during this study. The PerSC are grown during 14 days before adding the HSC for 5 days. The rationale behind the choice to explore PerSC's transcriptional phenotype and metabolism pathways at 7 days (Fig2g-d, Fig.3) and the HIF1a expression at day 3 (Fig4a) should be explain.

For example, niche markers are differentially expressed between d7 and d14 in the hypoxic conditions (NG2, CD46, CD51: d7 RNAseq = decreased expression in -gel hyp and glass hyp conditions and at d14 by FACS = increased expression) and LEPR expression in the +gel condition is decreased at the transcriptional level at d7 but unchanged by FACS analysis at d14. Are the immunomodulatory transcripts up regulated at 14 days as well?

In addition, the HSC are grown for 5 additional days, how is the phenotype of the PerSC at that time?

The gene expression of some of some key genes could be quantify at d14 as well as at the end of the culture with HSCs (in 0% and 100% conditions) to know to what environment the HSC are exposed.

Fig4 investigate the hypoxic status of the PerSC in the different bioengineered models.

The intensity of HIF1a in the quantification in 4a show an increase compared to the control condition, however the images presented have a very weak signal in the PEA/FN/BMP-2-gel and PEA/FN/BMP2 hyp. At the image processing step, the signal should be increased (in the same way for all the conditions) to display the staining so the reader can see it more clearly. A representative image of the control condition should be shown too. The authors conclude to a "prolonged nuclear presence of HIF1a", how is the nuclear expression of HIF1a at d7, d14 and at d19 after the HSC culture?

The aim of Fig5 is to investigate the role of nestin in HIF1a and metabolism regulations. The images associated to the quantifications in a, b, c should be shown. This also applies to the all the quantifications using immunofluorescence done in the supplemental figures.

Fig6 is looking at the HSC phenotype in the different models. It is clear that the +gel condition improve LT-HSC maintenance. But is it due to the nestin/HIF1a signalling described earlier? This could be achieved for example by seeding HSC on PerSC cells +/- nestin siRNA. In addition, it could be interesting to further characterize the LT-HSC vs ST-HSC phenotype present in the different conditions by adding more markers such as CD45R, CD90 and CD69f.

The ultimate goal of this new bioengineered niche model is to improve allogenic transplantation. To this end, it would be interesting to know how the HSC grown in the different model behave once they are transplanted in NSG mice.

Editing comments:

- In the main text and the legend, the reference to figure 1f and 1g are mixed.
- Fig4b: the legend boxes indicating the colour code for d7 and d14 is missing
- Statistics not in the right positions on Fig5e
- Histogram in Fig6c referred in the text as showing CD34+ CD38- population but on the graph legend it is CD34- CD38+

Reviewer #3 (Remarks to the Author):

This manuscript presents the development of a bone marrow-mimicking platform to culture hematopoietic stem cells (LT-HSC) in vitro. To mimic the desirable exterior layer of the stiff endosteal bone lining, the platform creates a multilayer of several components – 1) glass coverslip at the bottommost layer; 2) spin-casted poly(ethyl acrylate); 3) unfolded fibonectin fibrous network; 4) adsorbed BMP-2; and 5) 2D cultured perivascular stromal cells as a feeder components. Then, the platform was used to culture HSC in three difference conditions; 1) as it is under in vitro normoxia, 2) as it is under hypoxia, and 3) with collagen coverage on top under in vitro normoxia. Also, the cultures were compared under 0% GF media or 100% GF media. The authors found that the culture with collagen coverage on top under in vitro normoxia in 0% GF media was the best condition to maintain (but not expand) LT-HSC. The result of this manuscript can improve the maintenance of LT-HSC for allogenic transplantation to cure blood-related diseases.

After several rounds of reading and contemplating, the reviewer finds the content very interesting but also very niche and specific. For a better comprehension, the reviewer thinks the introduction should provide more general knowledge on HSC and bone marrow anatomy. An illustration of bone marrow anatomy (to compare with the experimental setup schematics) would be extremely helpful to covers a broad readership of Nature Communications beyond the biological field. The reviewer was confused in the first reading round about the differences between LT-HSC, ST-HSC and CD34+ve HSC, only to find out near the end of the result section that LT-HSC is CD34+ve HSC that survives through LTC-IC assay for more than 6 weeks. This definition of “LT-HSC” and “ST-HSC” based on 6 weeks threshold should be clarified right in the introduction, as well as whether it is a common standard or lab-specific standard (some papers even use 3-4 months as a cutting threshold). Moreover, the first impression after reading is that this manuscript aims to expand LT-HSC whereas the result shows that the platform could only maintain. Therefore, the narrative should be improved for better understanding for readers.

Therefore, the reviewer would like to recommend a major revision of the text (and probably the experiment design as well) on the following concerns. The reviewer is looking forward to reading a clearer version soon.

Major Concerns and Questions

- The hypoxic culture with collagen gel (+gel hypoxia) was absent in the manuscript. Since the authors showed that +gel condition did not cause hypoxic environment, the reviewer think +gel hypoxia condition should be included as well. In fact, it should be even more consistent to bone marrow microenvironment, isn't it?
- Many characterizations in this manuscript were done with fluorescent technique. Have the authors confirmed that PMA or PEA did not produce any background fluorescence? If so, please show in the supplements, or at least state in the method.
- It was not clear why BMP-2 attachment to FN is necessary. So far, the text only states that, “...fibrillar FN can efficiently bind and present BMP-2 ... to promote osteogenesis in MSCs in vitro.” Ref.35 showed the importance to “mouse” bone marrow mesenchymal progenitors, but is BMP-2 truly necessary for “human” PerSCs or HSCs? Have the authors tried culturing without BMP-2, or supplying BMP-2 in the media?
- What is the justification of using PerSCs from adipose tissue in this bone marrow modelling? Is it really substitutable? Also, the authors state that “...PerSCs could support HSC expansion” (page13, paragraph 1), but did not check whether culturing HSC on all bioengineered substrates without PerSCs could expand HSC population or not.

- Fig2cde: The caption states that the graphs show mean integrated intensity of marker as fold change to glass control. However, the fluorescent images showed that the cell number are different. Was that taken into account? If not, it means that the measured fold change might reflect the cell number rather than the expression intensity of individual cells. The control was PerSCs on glass, not on PEA/FN/BMP2, where the author justified that PerSCs cannot attach to PEA without protein interface. The reviewer thinks PEA substrates coated with poly-lysine or poly-ornithine should suffice for cell attachment without influencing much on PerSCs expression. Perhaps, the difference between glass control and PEA substrate might cause the difference in the observed cell number.

- Tukey test was used in almost all multiple comparisons. However, from my experience, comparing 4 groups already creates false positive (that is, what seems to be significantly different are not significantly different). This false positive occurrence is more likely to occur as the group number in multiple comparison gets larger. The reviewer suggests using Bonferroni multiple comparison test. Some useful information: <https://www.biostathandbook.com/multiplecomparisons.html> and <https://doi.org/10.4097/kja.d.18.00242>

- Figure 4d is really incomprehensible. Please improve this figure, comparing with two axes might be better (PC1 vs PC2 | PC2 vs PC3 | PC3 vs PC1).

o Additionally, the reviewer thinks it would be interesting to check the variable weight in each PC axis because it gives insights about the “main” metabolic variables that distinguish each condition.

- The authors suggests that nestin (which is abundant in +gel condition) helps improve cellular tolerance to oxidative stress by showing that cells in + gel conditions could survive through the exposure of H₂O₂. However, the method only states that “collagen gels were removed and 1000 μm (1 mM?) H₂O₂ added...” without detailing on the removal process (mechanical or enzymatic) or confirming a complete removal. The reviewer think it is also possible residue collagen fibers or remodeled proteins left on top of the cells help protect from H₂O₂, which is why including +gel hypoxia in the experiment is important. Although the author showed the results on Fig5e, it raised more question why (-gel -NES) samples is better than (+gel -NES) samples – in fact significance testing of (-gel -NES) with other samples are missing. The author should address these issues more clearly (or with additional experiments).

Minor Concern and Questions

- Fig1b: Please state whether these are AFM height image or phase images, and also show the color legend. Although these are phase images (from reviewer’s understanding), please so show height images in the supplement because phase images only show the relative differences in force interaction between AFM tip and the sample.

- Fig1f, g: The figures do not correspond to their captions and the text (page 3).

- Fig1d, e, g: What is the unit MFI? No explanation in the text or method was found. Also, why would total FN availability be lower than HFN7.1 availability?

- Fig1: Why are all bar graphs in Figure 1 evaluated with Mann-Whitney t-test for statistical significance? All the rest of figures in this manuscript was evaluated by Tukey test.
- Fig3d: The immunomodulatory transcript expression profile should not be split into two conditions. It should compare three conditions at the same time like Fig3c.
- Fig5c: The reviewer could not see the distribution profile of each set, whether it is normal, lognormal, or bimodal. Please set jitter format so that readers can see the distribution. Also, significance testing was missing especially with (-gel -NES) samples.
- Sup.Fig5: this supplement does not correspond to the text nor the caption
- Explanation about the current progress and gaps in human LT-HSC researches are missing. The reviewer finds these studies worth to be mentioned: <https://doi.org/10.1038/s41590-021-00925-1>, <https://doi.org/10.1016/j.stem.2021.07.003>

Reviewer #4 (Remarks to the Author):

In terms of significance, this manuscript argues that perivascular stromal cells (PerSCs) are a better stromal cell source to support the LT-HSC maintenance. The major focus of this paper is on optimizing conditions to enhance the survival, metabolism, and phenotype maintenance of the PerSCs. The data on how PerSCs supports LT-HSC and gene-edited HSCs is very thin and incomplete. The title is misleading.

When arguing PerSCs as a better source to support LT-HSC maintenance, discussion is completely missing on how this compares with other stromal cell types reported in the literature. Many studies in this area are more advanced (for example, “Mesenchymal stromal cells improve the transplantation outcome of CRISPR-Cas9 gene-edited human HSPCs” [10.1016/j.ymthe.2022.08.011](https://doi.org/10.1016/j.ymthe.2022.08.011)). Without such a comparison, it is hard to evaluate how significant this work is for the goal stated in the introduction. This study only used a stromal free “standard” culture as a control.

In terms of the results, most relevant data shown in Fig. 6 lack sufficient rigor and quantitative context as a support for the major conclusion. CFU improvement (Fig. 6d) probably does not pass significance test against the “gold standard” (this comparison is missing), based on the result shown. The results shown in Fig. 6g are not quantitative without replicates. Most importantly, the standard test for LT-HSC phenotype maintenance following expansion is in vivo engraftment assay. That data is missing from this manuscript.

In terms of novelty, this manuscript reports how culture conditions (substrate topography, 3D environment provided by the collagen hydrogel and metabolic conditioning, etc.) influence the survival and gene expression profile of PerSCs. While this may be new, the significance outside the context of HSC maintenance is unclear. More importantly, as a stromal layer, it is important to understand how PerSC cell state changes in the coculture condition, which is missing in the study or in discussion.

Rebuttal for Donnelly et al.

We apologise to the reviewers for the time it has taken us to resubmit the manuscript. We are grateful for the support for our work, suggestions and curiosities. We hope that they will see that we have taken all their concerns, comments and advice seriously and have performed many more experiments, some of which were long and complex. We are extremely grateful to the reviewers for their insight and we believe they have helped us to strengthen our work. We hope they feel our paper is now ready for publication in *Nature Communications*.

Reviewer 1.

We thank the reviewer for a comprehensive review that clearly articulates the novelty of the paper and recognises that our bioengineered system can be useful to maintain (or, potentially, expand) fully-functional HSCs for a source of clinical HSC transplantation or genetic editing of HSCs for treating hereditary blood disorders. The reviewer has a number of major and minor comments for us to address in order to improve our work. We have, therefore, performed a number of large experiments in response to their very helpful suggestions.

Major Comments:

A LT-HSC activity was evaluated only in vitro. Functional LT-HSC activity should be shown in vivo by xenotransplantation models.

While we strongly feel LTC-IC is a very good *in vitro* surrogate for mouse functional assays, we have carried out the *in vivo* work as directed by the reviewer. The data, shown in Figure 7f and supplementary Figure 13, have exceeded our expectation. The niche-maintained HSCs show engraftment potential similar to that of fresh HSC. However, the HSCs maintained in gold-standard media exhibited greatly reduced engraftment potential showing that during expansion, the cells lose the LT-HSC phenotype. Further, we increased our flow marker panel for the *in vitro* experiments (supplementary Figure 12). This is now described in the paper:

“To confirm this data, we first used extended flow cytometry analysis to identify CD45⁺ve/CD41⁻ve/CD16⁻ve/CD7⁻ve/CD38⁻ve/CD34⁺ve/CD90⁺ve/CD45RA⁻ve (LT-HSCs) from CD45⁺ve/CD41⁻ve/CD16⁻ve/CD7⁻ve/CD38⁻ve/CD34⁺ve/CD90⁻ve/CD45RA⁺ve (ST-HSCs) (Supplementary Figure 12a). This confirmed the 0% +gel niche retained LT-HSCs without expansion in the ST-HSC compartment observed in other conditions. Secondly, in vivo engraftment study showed that CD34⁺ve cells recovered from the niche after culture had engraftment potential similar to that of freshly thawed cells, demonstrating retention of the LT-HSC population (Figure 7f, supplementary Figure 13). It is notable that CD34⁺ve cells after culture in gold-standard media had significantly reduced engraftment potential (Figure 7f), alongside an increase in proportion of human MPPs to LT-HSCs in the bone marrow (supplementary Figure 13c). Together, this again demonstrates growth accompanied by differentiation in this control condition (Figure 7f, supplementary Figure 13).”

Supplementary Figure 12a, Gating strategy for extended flow cytometry panel comparing markers for LT-HSCs and ST-HSCs. Graph shows number of LT-HSC/ST-HSC per CD45⁺ cells and demonstrates expansion of the ST-HSC compartment in the gold standard, whereas PEA +gel maintains a population of LT-HSCs.

Figure 7f, HSCs were cultured for 5 days in gold standard or PEA +gel niches, or freshly thawed (fresh) immediately prior to sorting of CD45⁺/Lin^{ve}/CD34⁺ cells, and injected in irradiated mice. Blood of recipient mice was analysed at 6, 8, 10, 12 & 14 weeks (terminal bleed). Graph shows percentage of human donor-derived CD45⁺ cells. Number of recipient mice; fresh (positive control) = 2; PEA +gel = 3; gold standard = 5.

Supplementary Figure 1 | In vivo reconstitution of CD34⁺ cells from bioengineered niches. a, schematic shows experimental set up, briefly CD34⁺ cells were seeded in gold standard (GS) media with 100% cytokines, or in PEA/FN/BMP-2 +gel niches in 0% cytokine media. After 5 days CD45⁺/Lin⁻/CD34⁺ cells were sorted and transplanted into NOD-RAG-gc^{-/-} mice, to establish the lineage potential of human HSCs derived from indicated culture conditions in vivo. **b**, Representative gating strategy *i.* for analysis of % human CD45⁺ cells in peripheral blood, and *ii.* for stem cells marker analysis. **c**, Percentage human CD45⁺ cells in the BM and spleen, with comparison of % LT-HSCs and multipotent progenitors (MPPs) within the CD45⁺ population of the BM.

The factors that make the bioengineered system “endosteal niches for LT-HSCs” should be investigated and identified. e.g. What factors distinguished “endosteal niches for quiescence “ and “ the ones for proliferation or differentiation”?

Here, our ambition was to engineer an endosteal niche to preserve LT-HSCs. To achieve this, we overlaid a bone-like surface (PEA with fibronectin and BMP-2) with cultured PerSCs with a soft collagen hydrogel to mimic the stiffness of the bone marrow interface. We have modified the text in the introduction to more clearly articulate this:

“In this study, we employ bioengineering approaches in order to direct the physiology of the stromal feeder layer and enhance its ability to support and maintain quiescent LT-HSCs in culture without unwanted drift towards non-engrafting ST-HSCs and progenitor cells. In this study, we define LT-HSCs in vitro by survival in a long-term culture initiating cell assay (LTC-IC) and in vivo by engraftment post 3-months. For the bioengineered niche, we employ a polymeric surface that controls FN fibril formation allowing both an adhesive and direct GF signal to stromal cells.”

The paper has a strong focus on influencing the HSCs via the stromal cells, PerSCs, at the ‘endosteal’ surface and on defining the role of nestin in metabolism (HIF, glycolysis/OXPHOS), cytoprotection and the niche. We have extended this evaluation to look at nestin and HIF expression during HSC co-culture and shown that, in the bioengineered niches, these factors increase further when HSC are added:

“PerSC phenotype was next considered with respect to nestin and nuclear HIF1 α post-culture with CD34+ve cells, with and without gold-standard media. With use of gold-standard (100%) media in -gel and +gel conditions, only low levels of nestin and HIF1 α were observed (supplementary Figure 14) in PerSC phenotype. With use of 0% media, while low levels of nestin and HIF1 α were seen in -gel cultures, significant increases in both was noted in +gel cultures (supplementary Figure 14). This data indicates that the PerSCs maintain their niche phenotype best in the condition where best LT-HSC phenotype is maintained (+gel, 0%).”

Supplementary Figure 2 | PerSC phenotype after HSC co-culture at day 19. a, Nestin and b, HIF1 α co-localisation to the nucleus are both significantly increased in PerSCs during HSC co-culture in PEA/FN/BMP-2 +gel (PEA +gel) niches, only when 0% cytokine media is used (the LT-HSC supportive microenvironment). a, scale bar = 100 μ m, magenta = nestin, grey = actin, blue = DAPI. b, scale bar =

100 μm , dashed white line represents nuclear mask detected by DAPI staining, magenta = HIF1 α . $n = 3$ material replicates from 1 biological donor. Actin/DAPI outlines were used to measure individual cell/nuclei nestin/ HIF1 α integrated intensity with background correction.

“The gold standard condition”, in which a cocktail of several cytokines were just add to media, seems to be as good as, even better than, the bioengineered niche not only for maintaining quiescence (LT-HSC numbers in LTC-IC assay), but also for allowing their expansion. The authors should clarify the advantages for using the bioengineered system compared to conventional culture systems.

The reviewer’s guidance towards use of an *in vivo* engraftment model has really helped us to demonstrate the advantages as niche cultured HSCs engrafted as with fresh cells, while HSCs cultured in gold-standard media suffered very low engraftment (as per answer above).

Minor Comments:

In Fig1, the bioengineered niche on PEA showed several domains of FN exposed to bind integrins and trap growth factors. What growth factors for osteogenesis or niche functions were trapped in the structure?

We use the (heparin) growth factor binding domain of FN to load the surfaces with BMP-2 to help recreate the endosteal surface. With the reviewer’s help we have clarified this in the introduction text:

“The P5F3 domain of FN in particular has high binding affinity for several GF families⁴⁴. Here, we show that exogenous BMP-2 binds to PEA-FN (Figure 2g) allowing us to present this endosteal-related GF to cells. Although similar amounts of BMP-2 also bind to PMA, our previous work has shown that BMP-2 does not co-localise with FN on PMA, leading to its faster release^{35,36}. Thus, in this study, we use PEA to present FN and BMP-2 to recreate the milieu of the endosteal surface that supports HSC quiescence (maintenance) in our BM model.”

In Fig2c,d, quality of the immunofluorescence images are not enough to show individual cells. Images with higher magnification and resolution need to be shown.

New, clearer, images have replaced the old images as seen in Figure 3c and d.

Figure 3c, Representative images of changes in nestin expression evaluated by immunofluorescence at day 7 show increased expression in PEA/FN/BMP-2 +gel. Scale bar is 100 μm , magenta = nestin, cyan = DAPI. d, Representative images of vimentin expression, as evaluated by immunofluorescence

microscopy at day 7, shows no significant difference between models., scale bar is 100 μ m, magenta = vimentin, cyan = DAPI.

In Fig2e, this point is related to Fig1, higher amounts of SCF was observed in the PEA+gel condition. Is that due to mRNA expression, protein synthesis, degradation or secreted one trapped by ECM?

The SCF data is protein level data where we stopped export of SCF using brefeldin and performed image analysis. We now make this clear in the text:

“Next, PerSCs were treated with brefeldin A for the final 24 h of culture to prevent protein export and SCF levels were monitored by immunofluorescence. SCF production correlated with the observed increase in nestin expression in PEA +gel niches (Figure 3e, supplementary Figure 4c.”

How about other cytokines, cxcl12, SPP1, etc. essential for niche functions of PerSCs? What is the key factor that endows PerSCs with ability to support LT-HSCs?

The reviewer is clearly right that there are a range of HSC support factors that can be secreted by stromal cells, such as PerSCs. Here, we chose to look at SCF (as shown in Figure 3e, supplementary Figure 4c) and CXCL12 (as shown in supplementary Figure 4d). We focus on SCF as expression increased as we developed the PerSC niche (PerSCs are thought to be the main producer of SCF in the niche), while CXCL12 remained constant (CXCL12 is primarily expressed by CAR cells in the BM microenvironment (Cuotu et al, 2017 *Nature Biotechnology*). In our revision we have focussed on the ‘ultimate proof’ of the *in vivo* experiment and extending HSC flow cytometry, as directed by the reviewer.

In Fig2f, FACS analysis and the heatmap exhibit higher expression of the surface markers. Is the population expressing the markers higher or Is the cellular numbers expressing them increased? Dot plots and histograms will be useful to show this point.

This is an important point for clarity of data and we now provide all the data in new supplementary Figure 5.

Supplementary Figure 3 | Histograms of cell surface marker phenotyping. Corresponding to Figure 3f. **a**, CD51 and CD140a – pseudo-markers for nestin expression³ – are increased and show a double population (respectively) in nestin^{high} PerSCs in PEA/FN/BMP-2 +gel niches. **b**, shows representative histograms from one representative biological replicate for the heatmap in Figure 3f.

In Fig2h, although expression of “differentiation genes” are shown, functional assays such as in vitro tri-lineage differentiation have to be performed.

We have now performed the tri-lineage differentiation experiment to demonstrate multipotency as requested:

“We note that PerSCs have similar multipotency to MSCs (supplementary Figure 2).”

Supplementary Figure 4 | Trilineage differentiation potential of PerSCs. PerSCs were isolated and cultured \leq passage 5, then cultured in either control media or medias for osteogenic, chondrogenic and adipogenic differentiation of stromal stem cells. Histological stains were then used to assess differentiation after 4 wk culture. Alazarin red stains calcium deposits in osteogenic cultures. Safranin O is used to identify chondrocytes. Oil red O stains lipids in adipogenic cultures. Positive detection of all stains in differentiation medias demonstrates trilineage potential of PerSCs. Scale bar is 100 μ m.

In Fig3, the core data that characterize the differences among the three culture conditions have to be confirmed by qPCR.

For the majority of data we haven’t performed further validation at gene or protein level as we are looking at broad patterns rather than specific changes, and because RNAseq offers very high reliability that isn’t normally validated while microarray data always had to be. However, for key markers, such as nestin, SCF and CXCL12, we have confirmed data at the protein level in Figure 3 and supplementary Figure 4.

In Fig4a and Fig5d, the images have almost no information due to low quality and resolution.

We have improved the quality of these figures (now Figures 5a and 6d).

Figure 5a, Nuclear HIF1 α levels were compared at day 3 by immunofluorescence microscopy. HIF1 α levels increased in cells in all models, and significantly increased in +gel niches. Representative images are shown, magenta = HIF1 α , white shows nuclei outline based on DAPI staining, scale bar = 20 μ m. Graph shows means from 3 material replicates for 3 independent experiments with different donor cells. Each individual point represents 1 nuclei measurement with shape corresponding to mean.

Figure 6d, Nestin protects from oxidative stress. PerSCs in the bioengineered models were treated with 1 mM H₂O₂ to mimic oxidative stress. N= 3 material replicates, representative images shown, scale bar is 100 μ m, grey = Hoescht stained cells (live & dead); magenta = PI stained cells (dead).

In Fig6d,e, this point is raised as a major comment, please clarify the advantages for using the bioengineered system compared to conventional culture systems.

We have addressed this as directed by the reviewer above.

In Fig6g, CRISPR edited HSCs exhibited better survival in all of the three condition with 100% media. What is the biological differences between LT-HSCs and CRISPR edited HSCs?

The reviewer is, of course, correct that in this work we go as far as HSC survival in the niches post-CRISPR. We argue that this is a critical improvement towards enhancing success of gene editing HSCs as primary stem cells have notoriously poor transfection efficiency survival after using approaches such as electroporation, and so looking after each cell is key. The model provides a nice demonstrator of the possibility of using niches in this way. However, we recognise we have two limitations that we are currently working on. (1) The number of cells we put into CRISPR and recover from CRISPR and this means that we really need to move to lentiviral delivered CRISPR to scale up, and (2), the current degradability of the model that we are currently working to address using synthetic gels. These limitations mean that it is hard to provide the requested data on a reasonable time-scale, but it is something we are working towards. We hope the reviewer will accept this as out of scope for this current paper.

Reviewer 2.

The reviewer has provided a very insightful review that significantly helps us to improve our research. We are grateful that the reviewer finds our work to be of high quality and recognises the significance

of the work: “The acquisition and *in vitro* maintenance of functional haematopoietic stem cells (HSC) as primitive as possible for allogenic transplantation purposes is one of the current challenges in the field of haematopoiesis, therefore the presented work is very interesting and relevant to the field.” We are very happy that their guidance to undertake a number of new experiments has helped us to significantly improve and strengthen our work.

Major Comments:

The aim of Fig2 and 3 is to characterize PerSC cells phenotype on the +gel condition. From the images presented in Fig2.c it looks like the number of cells is increased compared to the other conditions. Even if the RNAseq data confirm the nestin overexpression in the +gel condition, the quantification of the images should take the number of cells in account. This could also be applied to Sup Fig1a. Same comment regarding vimentin expression quantification in figure 1d.

This is an important point and cell number was always taken into consideration for all quantitative analysis. We have made sure all the methods and legends state this and have specifically added to Figure 3 legend:

“Each point represents an individual field normalised to cell number.”

And to the methods:

“All images were background corrected and analysis normalised to cell number via DAPI/Hoescht staining, or individual cells measured, as stated in figure legends.”

Indeed, as expected, cell numbers were slightly reduced in the +gel conditions and so the representative images were misleading and so we have changed these now. We have also included a quantification of cell number to represent the general trend in supp figure 3b.

Supplementary Figure 5b, Cell number quantification, via Hoescht staining. Statistics by one-way ANOVA followed by Bonferroni multiple comparison test show no significant differences.

It looks like the PerSC cells didn't migrate inside the overlaid collagen hydrogel (from fig1 a scheme and Fig 6b doesn't show a CD45- population). Is it the case? If yes, is it an expected result? If PerSC do migrate inside, what is the proportion of PerSC cells found in the gel and in contact/proximity with HSC once co-cultured?

We have performed additional analysis to look at this and see very limited migration of PerSCs into the gel as is now shown in supplementary Figure 3a. Rather the HSCs have a tendency to migrate to

the bottom of the gels (near the PerSCs) as we can see with samples with and without the PerSCs added to the model (supplementary Figure 12b).

a

Supplementary Figure 6a. PerSC migration into collagen gels. Z projection of calcein stained PerSCs in PEA/FN/BMP-2 +gel niches. 50 µm slices show most PerSCs are within the 2000 µm of the well surface suggesting little upward migration through the gel.

b

Supplementary Figure 12b, HSC migration in collagen gels was assessed via NucRed staining of HSCs and distribution in the gels measured via z-projections at day 5. The histogram shows most HSCs migrate down toward the coverslip, regardless of the presence of PerSCs in the niche models.

Regarding the CD45^{-ve} population for the flow cytometry data in Figure 6 (now Figure 7). Due to the size differences of PerSCs and CD45^{+ve} cells, the FSC/SSC parameters for flow cytometry did not permit

detection of both populations in the same acquisition. Furthermore, enzymatic removal of PerSCs from the PEA systems required a different protocol than that used to recover CD45⁺ cells from the systems, as such the number of PerSCs would have been low in the samples analysed by flow cytometry in Figure 7.

It is unclear how the different time points were chosen during this study. The PerSC are grown during 14 days before adding the HSC for 5 days. The rationale behind the choice to explore PerSC's transcriptional phenotype and metabolism pathways at 7 days (Fig2g-d, Fig.3) and the HIF1 α expression at day 3 (Fig4a) should be explain. For example, niche markers are differentially expressed between d7 and d14 in the hypoxic conditions (NG2, CD46, CD51: d7 RNAseq = decreased expression in -gel hyp and glass hyp conditions and at d14 by FACS = increased expression) and LEPR expression in the +gel condition is decreased at the transcriptional level at d7 but unchanged by FACS analysis at d14. Are the immunomodulatory transcripts up regulated at 14 days as well?

For HIF1 α , while we looked at 3d, 7d and 14d (Figure 5, supplementary Figure 7a-c), we decided to focus on 3 days in the silencing experiments as we knew we were going to investigate HIF1 α relationship with nestin using siRNA. The siRNA is only effective for 72 hours, and PerSC transfection was more efficient in the niche systems if carried out before addition of the collagen gels (as the gels would limit the siRNA delivery to the cells).

For other experiments, we chose 7 days as an earlier time point to explore with RNAseq, and then observed sustained expression of key BM regulatory markers at day 14 and so decided to prime the cells in the niche for the longer time period.

In addition, the HSC are grown for 5 additional days, how is the phenotype of the PerSC at that time? The gene expression of some of some key genes could be quantify at d14 as well as at the end of the culture with HSCs (in 0% and 100% conditions) to know to what environment the HSC are exposed.

We have performed some new analysis of the PerSCs post-co-culture with HSCs looking at HIF1 α and nestin (supplementary Figure 14) showing that, in the bioengineered niches, these factors increase further when HSC are added:

“PerSC phenotype was next considered with respect to nestin and nuclear HIF1 α post-culture with CD34⁺ cells, with and without gold-standard media. With use of gold-standard (100%) media in -gel and +gel conditions, only low levels of nestin and HIF1 α were observed (supplementary Figure 14) in PerSC phenotype. With use of 0% media, while low levels of nestin and HIF1 α were seen in -gel cultures, significant increases in both was noted in +gel cultures (supplementary Figure 14). This data indicates that the PerSCs maintain their niche phenotype best in the condition where best LT-HSC phenotype is maintained (+gel, 0%).”

Supplementary Figure 7 | PerSC phenotype after HSC co-culture at day 19. a, Nestin and **b**, HIF1α co-localisation to the nucleus are both significantly increased in PerSCs during HSC co-culture in PEA/FN/BMP-2 +gel (PEA +gel) niches, only when 0% cytokine media is used (the LT-HSC supportive microenvironment). **a**, scale bar = 100 μm, magenta = nestin, grey = actin, blue = DAPI. **b**, scale bar = 100 μm, dashed white line represents nuclear mask detected by DAPI staining, magenta = HIF1α. n = 3 material replicates from 1 biological donor. Actin/DAPI outlines were used to measure individual cell/nuclei nestin/ HIF1α integrated intensity with background correction.

Fig4 investigate the hypoxic status of the PerSC in the different bioengineered models. The intensity of HIF1α in the quantification in 4a show an increase compared to the control condition, however the images presented have a very weak signal in the PEA/FN/BMP-2-gel and PEA/FN/BMP2 hyp. At the image processing step, the signal should be increased (in the same way for all the conditions) to display the staining so the reader can see it more clearly. A representative image of the control condition should be shown too. The authors conclude to a “prolonged nuclear presence of HIF1α”, how is the nuclear expression of HIF1α at d7, d14 and at d19 after the HSC culture?

We have changed the images in Figure 4a (now Figure 5a) as requested, and included representative images for control (glass) conditions. We have also quantified HIF1α at days 7 and 14 (supplementary Figure 7b&c) and after co-culture with HSCs (supplementary Figure 14, as above). The data demonstrates HIF1α is elevated in the niches at all time points as well as before and after co-culture.

Supplementary Figure 8. HIF1α co-localisation and downstream lactate regulation. a and b Nuclear HIF1α levels were compared at day 7 and 14 by immunofluorescence microscopy. HIF1α levels increased in cells in all models, and significantly increased in +gel niches. Each point represents 1 nuclei

measurement with shape corresponding to mean shape for 3 material replicates. Graph shows means \pm SEM.

The aim of Fig5 is to investigate the role of nestin in HIF1a and metabolism regulations. The images associated to the quantifications in a, b, c should be shown. This also applies to the all the quantifications using immunofluorescence done in the supplemental figures.

We have provided these images throughout the manuscript, specifically for Figures 3e/supplementary Figure 4c; Figure 6a/supplementary Figure 9; Figure 6b/supplementary Figure 10a; Figure 6c/supplementary Figure 10b; Figure 6e/supplementary Figure 10c; supplementary Figure 4d; and supplementary Figure 14).

Fig6 is looking at the HSC phenotype in the different models. It is clear that the +gel condition improve LT-HSC maintenance. But is it due to the nestin/HIF1a signalling described earlier? This could be achieved for example by seeding HSC on PerSC cells +/- nestin siRNA. In addition, it could be interesting to further characterize the LT-HSC vs ST-HSC phenotype present in the different conditions by adding more markers such as CD45R, CD90 and CD69f.

The reviewer asks what we feel is a great question that we are very interested in ourselves – what happens in the niche, to the HSCs, if we can silence nestin. We have attempted the experiment and have generated new data that indicates support of LT-HSC phenotype is reduced. However, we have too low-confidence in the data to include it in the paper, but share it with the reviewer. We have low-confidence because of the number of manipulations in removing and replacing the gels to allow us to knock-down nestin expression in PerSCs using siRNA and then the 72 hour limit of the silencing. To achieve this data, really we need a whole new approach. However, we hope this glance at very preliminary data is helpful to the reviewer (rebuttal Figure 1).

Rebuttal Figure 1. Nestin silencing effect on HSC co-culture. PerSCs were cultured for 14 days in niche models and on glass controls. At day 14, to ensure effective delivery of siRNA complexes to PerSCs, collagen gels were gently digested with collagenase (preliminary tests were performed to ensure this did not lead to removal of PerSCs from PEA/glass surfaces). siRNA complexes targeting NES or scrambled control RNA (scr) were then added to PerSCs and incubated for 6 hours. After incubation, transfection media was removed and new collagen gels were added to the PEA +gel niches and incubated to polymerise for 3 hours. Once gels had polymerised, CD34⁺ cells were added as per co-culture experiments in Figure 7. Due to the 72 h time limit of siRNA, CD45⁺ cells were harvested at day 3 and flow cytometry analysis of CD4/Lin/CD34/CD38 expression was performed. The % of CD45⁺/Lin⁻/CD34⁺/CD38⁻ cells represented as a fold change to the % of this population at day 0 is

shown. In all conditions there is a trend towards a decrease in the HSC population when NES is silenced compared to the scrambled control.

As directed, we have also included extended flow cytometry analysis to allow us to look more at LT/ST-HSC phenotype. The new data confirms that LT-HSC phenotype is better maintained in the bioengineered niches (supplementary Figure 12a).

“To confirm this data, we first used extended flow cytometry analysis to identify $CD45^{+ve}41a^{-ve}16^{-ve}7^{ve}38^{-ve}34^{+ve}90^{+ve}45RA^{-ve}$ (LT-HSCs) from $CD45^{+ve}41a^{-ve}16^{-ve}7^{-ve}38^{-ve}34^{+ve}90^{-ve}45RA^{+ve}$ (ST-HSCs) (supplementary Figure 12a). This confirmed the 0% +gel niche retained LT-HSCs without expansion in the ST-HSC compartment.”

Supplementary Figure 12 | LT-HSC and ST-HSC phenotyping by flow cytometry. a, Gating strategy for extended flow cytometry panel comparing markers for LT-HSCs and ST-HSCs. Graph shows number of LT-HSC/ST-HSC per CD45+ cells and demonstrates expansion of the ST-HSC compartment

in the gold standard, whereas PEA +gel maintains a population of LT-HSCs.

The ultimate goal of this new bioengineered niche model is to improve allogenic transplantation. To this end, it would be interesting to know how the HSC grown in the different model behave once they are transplanted in NSG mice.

We have undertaken the requested *in vivo* analysis and the data has exceeded our expectations. The niche-maintained HSCs show engraftment potential similar to that of fresh HSCs. However, the HSCs maintained in gold-standard media exhibited greatly reduced engraftment potential showing that during expansion, the cells lose the LT-HSC phenotype.

“Secondly, in vivo engraftment study showed that CD34⁺ cells recovered from the niche after culture had engraftment potential similar to that of freshly thawed cells, demonstrating retention of the LT-HSC population (Figure 7f, supplementary Figure 13). It is notable that CD34⁺ cells after culture in gold-standard media had significantly reduced engraftment potential which, again, demonstrates growth accompanied by differentiation in this control condition (Figure 7f, supplementary Figure 13).”

Figure 7f, HSCs were cultured for 5 days in gold standard or PEA +gel niches, or freshly thawed (fresh) immediately prior to sorting of CD45⁺/Lin⁻/CD34⁺ cells, and injected in irradiated mice. Blood of recipient mice was analysed at 6, 8, 10, 12 & 14 weeks (terminal bleed). Graph shows percentage of human donor-derived CD45⁺ cells. Number of recipient mice; fresh (positive control) = 2; PEA +gel = 3; gold standard = 5. See also supplementary Figure 13.

Supplementary Figure 9 | In vivo reconstitution of CD34⁺ve cells from bioengineered niches. **a**, schematic shows experimental set up, briefly CD34⁺ve cells were seeded in gold standard (GS) media with 100% cytokines, or in PEA/FN/BMP-2 +gel niches in 0% cytokine media. After 5 days CD45⁺ve/Lin⁺ve/CD34⁺ve cells were sorted and transplanted into NOD-RAG-gc^{-/-} mice, to establish the lineage potential of human HSCs derived from indicated culture conditions in vivo. **b**, Representative gating strategy *i.* for analysis of % human CD45⁺ve cells in peripheral blood, and *ii.* for stem cells marker analysis. **c**, Percentage human CD45⁺ve cells in the BM and spleen, with comparison of % LT-HSCs and multipotent progenitors (MPPs) within the CD45⁺ve population of the BM.

Editing comments

- In the main text and the legend, the reference to figure 1f and 1g are mixed.
- Fig4b: the legend boxes indicating the colour code for d7 and d14 is missing
- Statistics not in the right positions on Fig5e
- Histogram in Fig6c referred in the text as showing CD34+ CD38- population but on the graph legend it is CD34- CD38+

We thank the reviewer for noticing these and we have fixed them all.

Reviewer 3.

We thank the reviewer for their review which we feel has helped us to significantly improve our manuscript. They provide a nice overview of the work in their introductory statement, and ask for some clarifying edits to the introduction and to provide a schematic of the bone marrow niche, which we do in Figure 1. We have edited the final paragraph to provide clarity on our purpose and description of progenitors, ST-HSCs and LT-HSCs.

“In this study, we employ bioengineering approaches in order to direct the physiology of the stromal feeder layer and enhance its ability to support and maintain quiescent LT-HSC in culture without unwanted drift towards non-engrafting ST-HSCs and progenitor cells. In this study, we define LT-HSCs in vitro by survival in a long-term culture initiating cell assay (LTC-IC) and in vivo by engraftment post 3-month. For the bioengineered niche, we employ a polymeric surface that controls FN fibril formation allowing both an adhesive and direct GF signal to stromal cells^{35,36}, as well as a soft material to induce nestin expression to stimulate the stromal HSC-supportive phenotype^{12,13,18}. Our findings show that nestin expression, linked to changes in respiration, are central to enhanced LT-HSC maintenance. We further demonstrate that our bioengineered niches have the potential to support improved survival of CRISPR edited HSCs in this clinically important population.”

Figure 1. Long-term HSCs (LT-HSC) and Nestin^{high}/LEPR^{ve} Perivascular stem cells (PerSCs) reside close to the endosteum and the oxygenated arterioles that perfuse the endosteal niche, whereas short-term HSCs (ST-HSCs) and LEPR^{ve}/Nestin^{low} PerSCs are in close proximity to the sinusoidal vessels that carry

deoxygenated blood away from the BM. The BM extracellular matrix (ECM) is a low-stiffness network comprised primarily of collagen and fibronectin (FN). Modified with permissions.

Major Comments:

The hypoxic culture with collagen gel (+gel hypoxia) was absent in the manuscript. Since the authors showed that +gel condition did not cause hypoxic environment, the reviewer think +gel hypoxia condition should be included as well. In fact, it should be even more consistent to bone marrow microenvironment, isn't it?

While we agree with the reviewer that hypoxia is a niche factor, and, therefore, included control with hypoxic condition, our goal was to produce a niche that relied on bioengineering approaches alone – and we feel that we have achieved this goal. This is important if we wish to envisage use of such engineered niches in e.g. drug screening as easy-to-use, 96 well (or microfluidic) formats would be required to help adoption rather than adding in further complexity, such as the requirement for hypoxic workstations. We also note that the endosteal, or arteriolar, niche has higher oxygen tension than the perivascular niche. However, adding hypoxia is a next logical step to optimise the niches biologically to see if we can push beyond maintenance, as we achieve here, into areas such as HSC growth.

Many characterizations in this manuscript were done with fluorescent technique. Have the authors confirmed that PMA or PEA did not produce any background fluorescence? If so, please show in the supplements, or at least state in the method.

This is an important point as we did employ a lot of image analysis. All analysis included a background correction step (this has been added to the methods) and we now provide a supplementary Figure 1b that shows the (minimal) level of background that use of PMA and PEA added.

Supplementary Figure 1b. FN quantification. *b*, immunofluorescence analysis of PEA/PMA background fluorescence and FN quantification using in-cell western, demonstrates minimal background fluorescence observed in 700 channel containing no secondary antibodies, and 800 channel where there is no primary antibody target.

“All images were background corrected and analysis normalised to cell number via DAPI/Hoescht staining, or individual cells measured, as stated in figure legends.”

It was not clear why BMP-2 attachment to FN is necessary. So far, the text only states that, “...fibrillar FN can efficiently bind and present BMP-2 ... to promote osteogenesis in MSCs in vitro.” Ref.35 showed the importance to “mouse” bone marrow mesenchymal progenitors, but is BMP-2 truly necessary for “human” PerSCs or HSCs? Have the authors tried culturing without BMP-2, or supplying BMP-2 in the media?

We thank the reviewer for this point and do have some data that we have added into supplemental data that shows the enhancement to PerSC nestin expression and HIF1 α nuclear co-localisation with PEA/FN bound BMP-2 compared to addition of soluble BMP-2. Supplementary Figure 4a shows clearly that addition of a gel drives strong nestin expression. It also shows a trend of increase in nestin in PEA/FN/BMP-2 with no gel compared to glass +sol BMP-2. Supplementary Figure 7a again shows that PEA/FN/BMP-2 +gel drives the greatest HIF1 α nuclear co-localisation, but also shows that PEA/FN/BMP-2 -gel has significantly greater HIF1 α nuclear co-localisation compared to glass +sol BMP-2 control. These data, and the relevance of BMP-2 to the endosteal bone microenvironment, informed our decisions. We have added notes regarding these observations in the figure legends.

Supplementary Figure 4a, Nestin expression measured by immunofluorescence at day 14 of culture. Nestin expression significantly increased in perivascular stromal cells on the PEA/FN/BMP-2 system only in the presence of low stiffness gels (+gel niche). PMA is used as a control polymer and shows some increase with gel only by day 14. Glass with soluble BMP-2 (50 ng/ml) added to the media is used as a further control and shows the addition of soluble, not matrix bound, BMP-2 does not induce nestin expression. Graph shows mean integrated intensity as fold change to glass control \pm SEM, ****= $p < 0.001$, **= $p < 0.005$, determined by one-way ANOVA followed by Bonferroni's multiple comparison test. $N = 4$ material replicates. We note a small trend of increase as we switch BMP-2 from soluble (glass +sol. BMP-2) to solid-phase (PEA/FN/BMP-2 -gel).

Supplementary Figure 10a, HIF1a co-localisation to nucleus, comparison to additional controls PEA/FN +gel and glass +soluble BMP-2, shows PEA/FN/BMP-2 +gel leads to significantly higher levels of nuclear HIF1a at 3 days. Data from one experimental repeat that is included in Figure 5a, with 3 material replicates, statistics by one-way ANOVA with Bonferroni's multiple comparisons test. We note a significant increase as we switch BMP-2 from soluble (glass +sol. BMP-2) to solid-phase (PEA/FN/BMP-2 -gel).

What is the justification of using PerSCs from adipose tissue in this bone marrow modelling? Is it really substitutable? Also, the authors state that "...PerSCs could support HSC expansion" (page13, paragraph 1), but did not check whether culturing HSC on all bioengineered substrates without PerSCs could expand HSC population or not.

It is known from Dexter-type cultures that HSCs are better supported using stromal cells than when just placed in culture. Our work built on that of *Corselli et al* who showed that adipose PerSCs supported HSC phenotype better than bone marrow MSCs; we have now added this reference in as it is a clear omission to our logic in this part of the paper. *Corselli, M., et al. Perivascular support of human hematopoietic stem/progenitor cells. Blood 121, 2891-2901 (2013).*

Fig2cde: The caption states that the graphs show mean integrated intensity of marker as fold change to glass control. However, the fluorescent images showed that the cell number are different. Was that taken into account? If not, it means that the measured fold change might reflect the cell number rather than the expression intensity of individual cells. The control was PerSCs on glass, not on PEA/FN/BMP2, where the author justified that PerSCs cannot attach to PEA without protein interface. The reviewer thinks PEA substrates coated with poly-lysine or poly-ornithine should suffice for cell attachment without influencing much on PerSCs expression. Perhaps, the difference between glass control and PEA substrate might cause the difference in the observed cell number.

Normalisation of data is another important point and cell number was always taken into consideration for all quantitative analysis. We have made sure all the methods state this and have specifically added to Figure 3 legend:

“Each point represents an individual field normalised to cell number.”

We chose to use standard glass coverslips as a control as they are widely used in cell culture and because we were coating them with PEA. Therefore, they offer a very standard, non-bioengineered control. We also use PMA as a control in initial experiments. PMA is an appropriate control as it is chemically very similar to PEA, but it presents FN to the cells in globular conformation rather than in fibrillar conformation, as with PEA – and we show this to be important in the paper in supplementary Figure 4a and b.

Supplementary Figure 11a&b. a, Nestin expression measured by immunofluorescence at day 14 of culture. Nestin expression significantly increased in perivascular stromal cells on the PEA/FN/BMP-2 system only in the presence of low stiffness gels (+gel niche). PMA is used as a control polymer and shows some increase with gel only by day 14. Glass with soluble BMP-2 (50 ng/ml) added to the media is used as a further control and shows the addition of soluble, not matrix bound, BMP-2 does not induce nestin expression. Graph shows mean integrated intensity as fold change to glass control \pm SEM, ****= $p < 0.001$, **= $p < 0.005$, determined by one-way ANOVA followed by Bonferroni’s multiple comparison test. $N = 4$ material replicates. **b**, SCF production in niche models. Cells were cultured in niche systems for 14 days with brefeldin A treatment (5 μ g/ml in culture media) added for the final 24h to inhibit intracellular protein transport. Quantification of immunofluorescence staining with b, anti-SCF shows significant increase only in the PEA/FN/BMP-2 +gel condition. Graph shows mean \pm SEM, ****= $p < 0.001$, determined by one-way ANOVA followed by Bonferroni’s multiple comparison test. Each point represents integrated SCF intensity of 1ximage field normalised to cell number, from 4 material replicates.

Tukey test was used in almost all multiple comparisons. However, from my experience, comparing 4 groups already creates false positive (that is, what seems to be significantly different are not significantly different). This false positive occurrence is more likely to occur as the group number in

multiple comparison gets larger. The reviewer suggests using Bonferroni multiple comparison test. Some useful information: <https://www.biostathandbook.com/multiplecomparisons.html> and <https://doi.org/10.4097/kja.d.18.00242>

We have changed to Bonferroni test and maintain significance in the data as discussed in the paper (statistics are mentioned in figure legends and the methods section).

Figure 4d is really incomprehensible. Please improve this figure, comparing with two axes might be better (PC1 vs PC2 | PC2 vs PC3 | PC3 vs PC1).

We have changed the figure in line with the reviewer's recommendation and agree it is better (now Figure 5d).

Figure 5d, LC-MS metabolome analysis. Principal component analysis (PCA) of PEA/FN/BMP-2 models. PerSCs were cultured for 7 and 14 days. All detectable metabolites were subject to PCA. Each point represents 1 replicate (n= 3 or 4 material replicates). The PCA plot shows distinct clustering based on model and time point. PC1 separates both time points, whereas PC2 separates model condition. Hypoxic and +gel niches showed similar clustering in PC1.

Additionally, the reviewer thinks it would be interesting to check the variable weight in each PC axis because it gives insights about the “main” metabolic variables that distinguish each condition.

We now provide this in supplementary Figure 8.

a

Supplementary Figure 12 | Metabolic variables in principle components. *a*, Loading plots for PCA of all metabolites with most variable metabolites labelled. *b*, Scree plot shows variance for each principal component

The authors suggest that nestin (which is abundant in +gel condition) helps improve cellular tolerance to oxidative stress by showing that cells in +gel conditions could survive through the exposure of H₂O₂. However, the method only states that “collagen gels were removed and 1000 mm (1 mM?) H₂O₂ added...” without detailing on the removal process (mechanical or enzymatic) or confirming a complete removal. The reviewer thinks it is also possible residue collagen fibers or remodeled proteins left on top of the cells help protect from H₂O₂, which is why including +gel hypoxia in the experiment is important. Although the author showed the results on Fig5e, it raised more questions why (-gel -NES) samples are better than (+gel -NES) samples – in fact significance testing of (-gel -NES) with other samples are missing. The author should address these issues more clearly (or with additional experiments).

This is a good point that we had not addressed. Additional experiments show that the collagen removal is complete and, therefore, not a factor in the cytoprotective analysis. This is now shown in supplementary Figure 10c.

Supplementary Figure 10c, Collagen I staining on fixed (but not permeabilised) PEA +PerSC coverslips after 7 days culture. Fluorescence intensity demonstrates mechanical gel removal for cytotoxicity experiments is complete and does not leave residual collagen. PEA +collagen gel (without removal) was used as a positive control.

For the second part, the difference between -gel -NES and +gel -NES is only a trend and is not significantly different. We don't show all the statistics in 5e (now Figure 6e) as it is too much, but we provide the figure with full stats here to show that the -gel -NES condition is not better:

Rebuttal Figure 2. Figure 6e with all statistics shown.

Minor Comments:

Fig1b: Please state whether these are AFM height image or phase images, and also show the color legend. Although these are phase images (from reviewer's understanding), please so show height images in the supplement because phase images only show the relative differences in force interaction between AFM tip and the sample.

We apologise for not being clear. Figure 2 shows height images and we have now added a colour legend to Figure 2b. We now also include the phase images in supplementary Figure 1a as requested.

Figure 2b, AFM height images showing fibronectin (FN) spontaneously forms networks on PEA surfaces but not on PMA.

Supplementary Figure 13a, Phase images for AFM corresponding to Fig 2b.

Fig1f, g: The figures do not correspond to their captions and the text (page 3).

We thank the reviewer for noticing this and this has been corrected.

Fig1d, e, g: What is the unit MFI? No explanation in the text or method was found. Also, why would total FN availability be lower than HFN7.1 availability?

We have added to the axes that MFI is mean fluorescent intensity. As these are experiments performed with different kits and this is an arbitrary unit, arbitrary unit, one assessment (e.g. FN) cannot be compared to another (e.g. HFN7.1).

Fig1: Why are all bar graphs in Figure 1 evaluated with Mann-Whitney t-test for statistical significance? All the rest of figures in this manuscript was evaluated by Tukey test.

We chose to perform Mann-Whitney as we were performing too few comparisons to perform ANOVA and Tukey (or Bonferroni).

Fig3d: The immunomodulatory transcript expression profile should not be split into two conditions. It should compare three conditions at the same time like Fig3c.

We ran the analysis this way as we wanted to select out gene hits between the groups presented as, we believe, it makes the changes clearer to see.

Fig5c: The reviewer could not see the distribution profile of each set, whether it is normal, lognormal, or bimodal. Please set jitter format so that readers can see the distribution. Also, significance testing was missing especially with (-gel -NES) samples.

We have changes this as requested by the reviewer.

Sup.Fig5: this supplement does not correspond to the text nor the caption

This has been corrected.

Explanation about the current progress and gaps in human LT-HSC researches are missing. The reviewer finds these studies worth to be mentioned: <https://doi.org/10.1038/s41590-021-00925-1>, <https://doi.org/10.1016/j.stem.2021.07.003>

New biological insight, such as the work in these papers, can, indeed, help us develop next generation bioengineered niches by targeting particular HSC subpopulations and interacting with cells at the molecular level. We, therefore, add new discussion to the paper:

“Such advances in bioengineering will need to progress in hand with evolving understanding of LT-HSC quiescence and activation. For example, LT-HSCs, themselves, are heterogeneous and this could have practical importance in targeting specific populations of LT-HSCs with materials. CD112^{low} LT-HSCs, are differentially quiescence from CS122^{hi} LT-HSCs and better-preserved self-renewal and regeneration capacity under regenerative stress¹. Advancing biochemical information can also provide target pathways for bioengineers to design materials to interact with, as we do with integrins and GFs here. For example, transcription factor EB (TFEB) has been shown to provide lysosomal regulation to drive digestion of membrane receptors, such as transferrin and insulin receptors, in order to limit LT-HSC metabolism and prevent differentiation². The subcellular localization and activity of TFEB are regulated by mechanistic target of rapamycin (mTOR)-mediated phosphorylation, which occurs at the lysosomal surface³. Materials platforms have also been implicated in changes in chromatin organisation⁴⁻⁷ and, for LT-HSCs, understanding chromatin accessibility to suppress CCCTC-binding factor (CTCF), restrains LT-HSCs from transitioning to activated ST-HSCs⁸. These new biological insights guide us future bioengineered niche developments.”

Reviewer 4.

We are sorry that the reviewer found the work less interesting than our other reviewers did. We very much hope that in responding to their criticisms, we can highlight the main novelties of the work and convince the reviewer that the work is worthy of publishing at this high level.

They state that **“In terms of significance, this manuscript argues that perivascular stromal cells (PerSCs) are a better stromal cell source to support the LT-HSC maintenance”**. With respect, we would say that we argue that PerSCs, like MSCs, or perhaps, even, stromal fibroblasts, are a sensible stromal cell type to use in the models that we are building. They have some advantages in that they are a defined population selectable by FACS while ‘what is an MSC’ remains a moot point. There is also some literature that indicates that can offer better support that MSCs eg Corselli et al. However, we provide a justification to use them and argue some potential advantages that we feel are legitimate. However, this is not the focus of the paper, engineering a niche for LT-HSC maintenance is. Corselli, M., et al. *Perivascular support of human hematopoietic stem/progenitor cells. Blood* 121, 2891-2901 (2013).

The reviewer comments that **“the title is misleading”**. The title “Bioengineered niches that recreate physiological extracellular matrix organisation to support long-term haematopoietic stem cells”, we believe, accurately conveys our central paradigm – that FN organised by PEA within bioengineered models helps support maintenance of LT-HSC phenotype. We have added to the data on LT-HSC phenotype with new flow markers (supplementary Figure 12) and also with *in vivo* engraftment experiments (Figure 7f and supplementary Figure 13). We hope that this also helps with the reviewer’s comment that **“The data on how PerSCs supports LT-HSC and gene-edited HSCs is very thin and incomplete”**. We hope that it also helps with their comment that **“In terms of the results, most relevant data shown in Fig. 6 lack sufficient rigor and quantitative context as a support for the major conclusion. CFU improvement (Fig. 6d) probably does not pass significance test against the “gold standard” (this comparison is missing), based on the result shown. The results shown in Fig. 6g are not quantitative without replicates. Most importantly, the standard test for LT-HSC phenotype maintenance following expansion is *in vivo* engraftment assay. That data is missing from this manuscript”**.

Supplementary Figure 12. LT-HSC and ST-HSC phenotyping by flow cytometry. *a*, Gating strategy for extended flow cytometry panel comparing markers for LT-HSCs and ST-HSCs. Graph shows number of LT-HSC/ST-HSC per CD45⁺ cells and demonstrates expansion of the ST-HSC compartment in the gold standard, whereas PEA +gel maintains a population of LT-HSCs.

Figure 7f, HSCs were cultured for 5 days in gold standard or PEA +gel niches, or freshly thawed (fresh) immediately prior to sorting of CD45⁺ve/Lin⁻ve/CD34⁺ve cells, and injected in irradiated mice. Blood of

recipient mice was analysed at 6, 8, 10, 12 & 14 weeks (terminal bleed). Graph shows percentage of human donor-derived CD45⁺ cells. Number of recipient mice; fresh (positive control) = 2; PEA +gel = 3; gold standard = 5. See also supplementary Figure 13.

Supplementary Figure 14 | *In vivo* reconstitution of CD34⁺ cells from bioengineered niches. **a**, schematic shows experimental set up, briefly CD34⁺ cells were seeded in gold standard (GS) media with 100% cytokines, or in PEA/FN/BMP-2 +gel niches in 0% cytokine media. After 5 days CD45⁺/Lin⁻/CD34⁺ cells were sorted and transplanted into NOD-RAG-gc^{-/-} mice, to establish the lineage potential of human HSCs derived from indicated culture conditions *in vivo*. **b**, Representative gating strategy i. for analysis of % human CD45⁺ cells in peripheral blood, and ii. for stem cells marker analysis. **c**, Percentage human CD45⁺ cells in the BM and spleen, with comparison of % LT-HSCs and multipotent progenitors (MPPs) within the CD45⁺ population of the BM.

The reviewer then states that **“When arguing PerSCs as a better source to support LT-HSC maintenance, discussion is completely missing on how this compares with other stromal cell types reported in the literature. Many studies in this area are more advanced (for example, “Mesenchymal stromal cells improve the transplantation outcome of CRISPR-Cas9 gene-edited human HSPCs” 10.1016/j.ymthe.2022.08.011). Without such a comparison, it is hard to evaluate how significant this work is for the goal stated in the introduction. This study only used a stromal free “standard” culture as a control”**.

The paper provided is very nice (and we have added it to our manuscript in the discussion section) and the reviewer is clearly correct that it goes way beyond our work on the CRISPR edited HSCs. However, our focus, at this time, is to take us to the point we have a bioengineered system that provides better LT-HSC maintenance and to provide a demonstrator of potential utility – here using CRISPR as a strategy many HSC biologists are interested in (as evidenced by the reference suggested).

We do provide a relevant control as we optimise the PerSC cultures, namely PMA with FN and PerSCs; here compare globular (PMA) and fibrillar (PEA) FN and show that PEA-FN-PerSCs is the optimal condition to take through to LT-HSC analysis. Our flow, LTC-IC and *in vivo* data show that we meet our objective.

Further, caution has to be considered when considering expansion of HSC, normal or gene edited. The paper provided focusses on expanding the gene edited-HSC population using Dexter-like culture. Historically, the result tends to be that expansion is in the ST-HSC population rather than LT-HSC (as illustrated by our LTC-IC and *in vivo* work) and so even promising initial clinical results, as has been seen in trials of PGE2 and SP1, can ultimately lead to lack of long-term engraftment. Therefore, we believe our strategy to study maintenance *ex vivo* is a good one. Further, we believe that focussing on survival post-CRISPR to maximise GE-HSC number is also a good one. Indeed, we employ NHEJ that enables us to target the quiescent LT-HSC population more specifically and we believe this is a useful approach to the field.

“Recent work shows that stromal support for gene edited-HSCs (Dexter-like cultures) provides required HSC support factors and can provide better engraftment⁶⁶. Bioengineered niches can be envisaged to offer further enhanced control of gene edited-LT-HSC phenotype.”

The reviewer finally comments that **“In terms of novelty, this manuscript reports how culture conditions (substrate topography, 3D environment provided by the collagen hydrogel and metabolic conditioning, etc.) influence the survival and gene expression profile of PerSCs. While this may be new, the significance outside the context of HSC maintenance is unclear. More importantly, as a stromal layer, it is important to understand how PerSC cell state changes in the coculture condition, which is missing in the study or in discussion”**.

We have now also quantified HIF1 α at days 7 and 14 (supplementary Figure 7b&c) and HIF1 α and nestin expression after co-culture with HSCs (supplementary Figure 14). The data demonstrates HIF1 α and nestin are elevated in the niches at all time points as well as before and after co-culture.

Supplementary Figure 15a & b, Nuclear HIF1 α levels were compared at day 7 and 14 by immunofluorescence microscopy. HIF1 α levels increased in cells in all models, and significantly increased in +gel niches. Each point represents 1 nuclei measurement with shape corresponding to mean shape for 3 material replicates. Graph shows means \pm SEM.

Supplementary Figure 16 | PerSC phenotype after HSC co-culture at day 19. **a**, Nestin and **b**, HIF1 α co-localisation to the nucleus are both significantly increased in PerSCs during HSC co-culture in PEA/FN/BMP-2 +gel (PEA +gel) niches, only when 0% cytokine media is used (the LT-HSC supportive microenvironment). **a**, scale bar = 100 μ m, magenta = nestin, grey = actin, blue = DAPI. **b**, scale bar = 100 μ m, dashed white line represents nuclear mask detected by DAPI staining, magenta = HIF1 α . $n = 3$ material replicates from 1 biological donor. Actin/DAPI outlines were used to measure individual cell/nuclei nestin/ HIF1 α integrated intensity with background correction.

REVIEWERS' COMMENTS

Reviewer #1 (Remarks to the Author):

The authors have conducted the experiments suggested by this reviewer.

This article is now acceptable to be published in nature communications, although there are some mislabeling and typo remained in the text. Please correct them.

Reviewer #3 (Remarks to the Author):

The reviewer finds this revised manuscript more logical and clarified. The addition of transplantation results really confirms the claim of this research that the bioengineered construct can maintain LT-HSC activity in vitro. However, there are some minor issues in this manuscript that needs to be corrected before publication. Please see the attached files.

Reviewer #4 (Remarks to the Author):

Overall, this revised manuscript is markedly improved over the original submission. Particularly the new data added on functional LT-HSC in vivo data, careful characterization of the PerSC phenotype change during the culture, and benchmark discussion, provide more support to the statements made in the manuscript. Given these improvements, the comments from this reviewer relating to the significance, title description, data support on PerSCs and LT-HSC functional assessment, are addressed adequately.

Relating to the “Benchmark comparison with other work in the literature”, the modification by the author in discussion section is acceptable.

Relating to the “novelty”, the new data added on PerSCs identity and function provide the utility support, adding to the support for novelty of PerSC in creating an effective HSC niche.

I would recommend the acceptance of this manuscript in its current form.

Reviewer #5 (Remarks to the Author):

The revised version of the manuscript successfully shed light on the questions and remarks that was raised during the first review process. Thus, I recommend the publication of the manuscript in Nature Communications without further review

The reviewer finds this revised manuscript more logical and clarified. The addition of transplantation results really confirms the claim of this research that the bioengineered construct can maintain LT-HSC activity in vitro. However, there are some minor issues in this manuscript that needs to be corrected before publication.

Minor Concern

- **Figure 2b:** The reviewer checked the authors' previous work on Advanced Science 2019 <https://doi.org/10.1002/adv.201800361>, and found that the previously reported fibronectin network (Figure 1e, Figure 3S on Cheng et al. Adv Sci. 2019) were much thinner and smaller (almost 2 times) than the fibronectin network in this manuscript (Figure 2b). When comparing the experimental setup and method, both works follows the same protocol – 200ul of 20ug/ml FN in PBS dropped on spin-coated/casted PEA disc; 10 min adsorption; 2x wash with PBS/DPBS; 1x wash with DI water under N₂ stream. AFM setup is word-by-word exactly the same. The author needs to recheck and explain the inconsistency here. The author might want to clarify if PBS they used contained calcium and magnesium or not.
- **Figure 2b vs Figure S1a:** Why are the phase images rotated by 90 degrees compared to the corresponding height images?
- **Figure S1b:** The reviewer understood the background measurement with no secondary antibodies (700 channel) but do not understand what happens in the 800 channel. What does it mean by “no primary antibody target”? Did the authors stain with secondary antibody and no primary antibody? Did the secondary antibody preferentially bind to +FN condition? Or were they the sample? Why did the rightmost PMA +FN expressed almost minimal fluorescence?
- **Figure 5b:** Please showed the datapoint distribution in this bar graph.
- **Page 6 & Figure S4a:** The reviewer would like to thank the authors in investigating further on this matter per the reviewer's previous comment.
 - However, the authors stated that in PEA/FN/BMP-2 +gel condition, nestin was upregulated. By contrast, nestin was expressed at a basal level in PEA/FN/BMP-2-gel, PEA/FN/BMP-2 hypoxic, PMA/FN/BMP-2 -gel, and PMA/FN/BMP-2 +gel. Yet, Figure S4a did not show a direct comparison between (1) PEA/FN/BMP-2 +gel vs PMA/FN/BMP-2 +gel, and (2) PEA/FN/BMP-2 +gel vs PMA/FN/BMP-2 -gel. Therefore, the authors should not overstate that PMA/FN/BMP-2 ± gel expressed a basal level of nestin without showing a direct statistical comparison, especially when the authors clearly stated in the caption that PMA is used as a control polymer.
 - The significance mark among PEA/FN/BMP-2 +gel, PEA/FN/BMP-2 hypoxia and PEA/FN - gel is missing.
- **Page 10 & Figure 8b:** Thank you so much for splitting 3D PCA into 2D PCAs. This really tells that whatever variable that contribute the most to PC2 helps separate 7day data from 14day data. Likewise whatever variable that contribute the most to PC1 distinguishes +gel, - gel, and hypoxia conditions from one another. But when looking into the Figure S8a, it is quite hard to understand which variable contribute the most, especially when there is no discussion on these results in the text. It is nice to have, but not really necessary to have. Also, he author might need to elaborate more to transition from LC-MS/PCA to focusing on glycolysis and TCA cycle.
- **Figure S4b:** Significance mark (****) is missing
- **Figure 7f:** Why does one PEA+gel curve remain flat throughout the duration course?

We thank our reviewers for their positive comments and reviewers 1,4, and 5 for their decision of accept and reviewer 3 for their minor comments to help us to improve our paper further.

Reviewer 3

Minor Concern

- **Figure 2b:** The reviewer checked the authors' previous work on Advanced Science 2019 <https://doi.org/10.1002/advs.201800361>, and found that the previously reported fibronectin network (Figure 1e, Figure 3S on Cheng et al. Adv Sci. 2019) were much thinner and smaller (almost 2 times) than the fibronectin network in this manuscript (Figure 2b). When comparing the experimental setup and method, both works follows the same protocol – 200ul of 20ug/ml FN in PBS dropped on spin-coated/casted PEA disc; 10 min adsorption; 2x wash with PBS/DPBS; 1x wash with DI water under N2 stream. AFM setup is word-by-word exactly the same. The author needs to recheck and explain the inconsistency here. The author might want to clarify if PBS they used contained calcium and magnesium or not.

The PBS contained calcium and magnesium and the methods have been updated to state this. In our previous work using PEA coatings to form FN networks, there are some key differences in aims and application; the Cheng et al paper looks primarily at a different method of PEA polymerisation (plasma coating), and uses spin coated PEA only as a control. As PEA is nonbiodegradable, the aim of the work was to produce a layer of PEA that is thin enough for the body to metabolise (<10s of nm) when a scaffold is coated and implanted for bone regeneration applications. As such, the spin coated samples were prepared with a 4% wt/wt PEA/toluene solution and spun at 3000 rpm to produce as thin a layer as possible using the control spin coating method. In the work presented in this manuscript, we aimed to engineer in vitro models for long-term culture of stromal stem cells and HSCs, therefore a thicker more robust layer of the polymer was required to avoid coating loss during culture. Here the PEA was prepared at 6 and 9% wt/wt with toluene (dependent on batch), and spun at 2000 rpm, this difference in PEA thickness and polymerisation/coating approach should explain the difference observed by the reviewer.

- **Figure 2b vs Figure S1a:** Why are the phase images rotated by 90 degrees compared to the corresponding height images?

We thank the reviewer for noting this. We have rotated the images in Figure 2, they are now correct in both Figure 2 and supplementary Figure 1.

- **Figure S1b:** The reviewer understood the background measurement with no secondary antibodies (700 channel) but do not understand what happens in the 800 channel. What does it mean by “no primary antibody target”? Did the authors stain with secondary antibody and no primary antibody? Did the secondary antibody preferentially bind to +FN condition? Or were they the sample? Why did the rightmost PMA +FN expressed almost minimal fluorescence?

We have corrected the figure legend to make this clearer. No primary target refers to the materials samples that are not coated with FN, and have subsequently been stained with an anti-FN antibody. The rightmost PMA+FN sample demonstrates some variation in samples.

- **Figure 5b:** Please showed the datapoint distribution in this bar graph.

This has been corrected.

- **Page 6 & Figure S4a:** The reviewer would like to thank the authors in investigating further on this matter per the reviewer's previous comment.
 - However, the authors stated that in PEA/FN/BMP-2 +gel condition, nestin was upregulated. By contrast, nestin was expressed at a basal level in PEA/FN/BMP-2-gel, PEA/FN/BMP-2 hypoxic, PMA/FN/BMP-2 -gel, and PMA/FN/BMP-2 +gel. Yet, Figure S4a did not show a direct comparison between (1) PEA/FN/BMP-2 +gel **vs** PMA/FN/BMP-2 +gel, and (2) PEA/FN/BMP-2 +gel **vs** PMA/FN/BMP-2 -gel. Therefore, the authors should not overstate that PMA/FN/BMP-2 ± gel expressed a basal level of nestin without showing a direct statistical comparison, especially when the authors clearly stated in the caption that PMA is used as a control polymer.
 - The significance mark among PEA/FN/BMP-2 +gel, PEA/FN/BMP-2 hypoxia and PEA/FN - gel is missing.

These statics have now been added to supplementary Figure 4.

- **Page 10 & Figure 8b:** Thank you so much for splitting 3D PCA into 2D PCAs. This really tells that whatever variable that contribute the most to PC2 helps separate 7day data from 14day data. Likewise whatever variable that contribute the most to PC1 distinguishes +gel, - gel, and hypoxia conditions from one another. But when looking into the Figure S8a, it is quite hard to understand which variable contribute the most, especially when there is no discussion on these results in the text. It is nice to have, but not really necessary to have. Also, he author might need to elaborate more to transition from LC-MS/PCA to focusing on glycolysis and TCA cycle.

We are glad the reviewer agrees the 2D PCA plots are more informative. Regarding the transition from LC-MS/PCA to the focus on glycolysis and TCA cycle, we begin this section investigating hypoxia-related behaviour and metabolism – HIF1a, and downstream effects – the PCA is to show overall differences between the metabolism of each condition, the following focus on glycolysis/metabolism is explained in the beginning of the results section relating to Figure 5:

'Metabolic regulation in the BM niche

In order to elucidate further the role of hypoxia in the BM niche, we investigated the role of hypoxia and stiffness on stromal niche phenotypes by using our modular platform to compare the effects of hypoxia in our bioengineered models. HIF1 α is a master regulator of metabolism. It regulates both glycolysis and oxidative phosphorylation, including anaerobic glycolysis⁵³.'

- **Figure S4b:** Significance mark (****) is missing

This has been corrected, we thank the reviewer for pointing this out.

- **Figure 7f:** Why does one PEA+gel curve remain flat throughout the duration course?

We thank the reviewer for this comment. Human cells don't always engraft into animal models and this can be very clearly seen for HSCs cultured in gold standard media before introducing to the animal. For the HSCs cultured in the niches, there was one animal that did not engraft the cells – however, the other animals engrafted the cells as well as freshly isolated HSCs. We have now commented on this in the text.